

# Evolution of Antarctic firn air content under three future warming scenarios

Sanne B.M. Veldhuijsen[1], Willem Jan van de Berg[1], Peter Kuipers Munneke[1], and Michiel R. van den Broeke[1]

[1]Institute for Marine and Atmospheric Research Utrecht, Utrecht University, Utrecht, The Netherlands

**Correspondence:** Sanne B.M. Veldhuijsen (s.b.m.veldhuijsen@uu.nl)

**Abstract.** The Antarctic firn layer provides pore space in which an estimated 94 to 96 % of the surface melt refreezes or is retained as liquid water. Future depletion of pore space in the firn layer by increased surface melt, densification rates and formation of impermeable ice slabs can potentially lead to extensive meltwater ponding, followed by ice-shelf disintegration by hydrofracturing. Here, we investigate 21st century evolution of the total firn air content (FAC) and accessible FAC (i.e. the pore space that is accessible for meltwater) across Antarctica. We use the semi-empirical firn model IMAU-FDM with
an updated dynamical densification expression. The firn model is forced by general circulation model CESM2 output for three climate scenarios (SSP1-2.6, SSP2-4.5 and SSP5-8.5), dynamically downscaled to a 27 km horizontal resolution by the regional climate model RACMO2.3p2. To estimate the accessible FAC, we prescribe a relationship between ice-slab thickness and permeability. In our simulations, ice shelves in the Antarctic Peninsula and Roi Baudouin ice shelf in Dronning Maud
Land are particularly vulnerable to FAC depletion (> 50 % decrease), even for strong and intermediate mitigation scenarios. Especially in the high-end warming scenario, the formation of ice slabs further reduces accessible FAC on ice shelves with low accumulation rates (current rates of $< 500\,\mathrm{mm\,yr^{-1}}$), including many East-Antarctic ice shelves and on Filchner-Ronne, Ross, Pine Island and Larsen C ice shelves. Our results underline the different response of low- and high-accumulation ice shelves to atmospheric warming, indicating a potentially large impact of ice slab formation on the viability of low-accumulation ice
shelves.

## 1 Introduction

The Antarctic ice sheet (AIS) has been losing mass since at least 2002 (Shepherd et al., 2018; Rignot et al., 2019), contributing to around 10 % of global average sea level rise since 1993 (Oppenheimer et al., 2019). This mass loss is mainly driven by enhanced iceberg calving and basal melting beneath ice shelves (Smith et al., 2020). Both reduce their buttressing effect,
allowing tributary glaciers to accelerate, increasing ice discharge into the ocean. On the Antarctic Peninsula, the warmest region of Antarctica, mass loss is also driven by surface melt. Here, extensive melt has led to meltwater ponding and subsequently to ice-shelf disintegration by hydrofracturing, after which mass loss from tributary glaciers has accelerated (Rignot et al., 2004; Banwell et al., 2013).



Not all ice shelves are susceptible to meltwater induced hydrofracturing. Firstly, extensive surface melt typically only leads to
meltwater ponding when the firn layer lacks sufficient pore space for meltwater to percolate downward and refreeze. Currently,
an estimated 94 to 96 % of the surface melt is retained within the firn (Medley et al., 2022; Van Wessem et al., 2018). Secondly,
meltwater induced hydrofracturing also requires sufficient tensile stress. Hydrofracturing, in turn, only induces mass loss if
the ice shelf provides substantial buttressing. Currently, 60 % of the ice shelves (by area) are vulnerable to hydrofracturing if
inundated by meltwater (i.e. where sufficient tensile stress is present), and buttress upstream ice (Lai et al., 2020). Hence, to
assess the future stability of ice shelves and predict mass loss from the AIS, it is important to estimate the future evolution of
the AIS firn layer.

Under future warming, we anticipate more surface melt, faster firn densification and increased formation of impermeable
ice slabs (i.e. ice layers > 1 m thick) by refreezing (Ligtenberg et al., 2014; MacFerrin et al., 2019; Kittel et al., 2021). These
processes deplete the firn air content (FAC) and consequently accelerate firn saturation and ponding by melt water. On the other
hand, snowfall is projected to increase as well (Kittel et al., 2021), adding additional pore space to the firn. Climate models
have been used to assess the impacts of future climatic changes on Antarctica's firn saturation. van Wessem et al. (2023) assess
future melt ponding on ice shelves based on the exceedance of a melt-over-accumulation ratio (MOA) of 0.7, in a diagnostic
study that does not explicitly consider the firn layer itself. Gilbert and Kittel (2021) use runoff and melt from climate models
as indicators of ice-shelf instability. While runoff is a measure of firn saturation, the snow surface schemes of climate models
have a limited vertical resolution and are not optimized to represent the firn layer and its physical processes in detail. The
main advantage of using an offline firn model instead is the lower computational cost, which enables it to use a higher vertical
resolution, a proper initialization of the firn layer and to perform extensive sensitivity tests. The disadvantage of using an offline
firn model is that interaction with the atmosphere is not possible.

Firn models forced by outputs of regional climate models or reanalysis datasets, have been used to simulate the current
(1979 till present) AIS firn layer (Gardner et al., 2023; Keenan et al., 2021; Medley et al., 2022; Veldhuijsen et al., 2023a). Firn
models have also been forced by outputs of regional climate models to simulate FAC evolution in response to climate change
scenarios (Ligtenberg et al., 2014; Munneke et al., 2014). However, the densification equations used in firn models are often
based on assumptions of constant accumulation and temperature (Arthern et al., 2010), which are invalid for the projected
transient climate of the 21st century. Simulating future FAC evolution requires firn densification expressions that allow for
changing climatic conditions. Moreover, using only total FAC to assess the firn's meltwater buffering capacity overlooks the
impact of near-surface ice slabs formed by meltwater refreezing. These ice slabs, which are common in Greenland (MacFerrin
et al., 2019; Culberg et al., 2021) and have locally been observed in Antarctica on Larsen C ice shelf (Hubbard et al., 2016),
can impede vertical meltwater percolation to deeper firn, limiting the fraction of the FAC that is accessible for meltwater. To
assess the meltwater buffering capacity of firn, it is therefore important not only to consider total FAC, but also to include the
impact of ice slabs, thereby considering the FAC that is accessible for meltwater (i.e. the accessible FAC).

Here, we use the IMAU Firn Densification Model (IMAU-FDM) for Antarctica, which has previously been evaluated for
the contemporary climate (1979-2020) (Veldhuijsen et al., 2023a). IMAU-FDM is driven by realizations of CESM2 of the
scenarios SSP1-2.6, SSP2-4.5 and SSP5-8.5 for the period 1950-2100, dynamically downscaled to 27 km resolution with



RACMO2.3p2. To allow for changing climatic conditions, we updated and evaluated the densification equation (Sections 2 and 3). In Sections 4 and 5, we present and discuss the response of the AIS firn layer to future warming scenarios and specifically focus on the evolution of total FAC, accessible FAC and runoff. We finish with conclusions in Section 6.

## 2 Methods

### 2.1 IMAU-FDM

IMAU-FDM is a semi-empirical 1D firn densification model that simulates the evolution of firn density, temperature, liquid water content and surface height changes due to firn and surface mass balance (SMB) and energy balance (SEB) processes. Firn compaction is calculated based on the semi-empirical dry-snow densification equations of Arthern et al. (2010), discussed in more detail below. The conduction of heat is simulated by using a one-dimensional heat transfer equation, which couples vertical heat conduction to temperature gradients through the thermal conductivity of firn. The thermal conductivity is computed as a function of temperature and density. Meltwater percolation is simulated using the bucket method, whereby each firn layer has a maximum irreducible water content that decreases with increasing density (Coléou et al., 1999). The meltwater can percolate through all layers in a single time step, and (partly) refreezes when it reaches a layer with a temperature below the freezing point. Standing water and lateral runoff over ice layers are currently not considered. An equilibrium initial firn column is obtained by looping over a reference climate until the entire firn column is refreshed. For further details of the model setup we refer to Veldhuijsen et al. (2023a) and Brils et al. (2022). Version v1.2A of IMAU-FDM (referred as FDM v1.2A) has been extensively evaluated over Antarctica against in situ observations of firn density and temperature, and remote sensing altimetry measurements (Veldhuijsen et al., 2023a). In this study we update the model to version IMAU-FDM v1.2AD (referred to as FDM v1.2AD) by implementing a densification expression that is suitable for transient climate change experiments, as is described in the next sections.

#### 2.1.1 General densification expressions

As discussed in Arthern et al. (2010), the evolution equations for density ($\rho$), squared grain size ($r^2$), and overburden pressure ($\sigma$) for dry snow are given by:

$$\frac{\mathrm{d}\rho}{\mathrm{d}t} = k_c(\rho_i - \rho)e^{\left(\frac{E_c}{RT}\right)}\frac{\sigma}{r^2} \tag{1}$$

$$\frac{\mathrm{d}r^2}{\mathrm{d}t} = k_g e^{\left(-\frac{E_g}{RT}\right)} \tag{2}$$

$$\frac{\mathrm{d}\sigma}{\mathrm{d}t} = g\dot{b}_{inst} \tag{3}$$





where $k_c$ and $k_g$ are constants, $\rho_i$ is the density of bubble free ice ($917 \ \mathrm{kg \, m^{-3}}$), $\rho$ is the layer density ($\mathrm{kg \, m^{-3}}$), $T$ is the instantaneous layer temperature (K), $R$ is the universal gas constant, $g$ is the gravitational acceleration, $\dot{b}_{inst}$ the instantaneous accumulation rate ($\mathrm{kg \, m^{-2} \, s}$), and $\mathrm{E}_c$ and $\mathrm{E}_g$ are the activation energies for creep ($60.0 \ \mathrm{kJ \, mol^{-1}}$) and grain growth ($42.4$

$\mathrm{kJ \, mol^{-1}}$), respectively. Assuming negligible initial grain size and a constant temperature and accumulation in Eqs. (2) and (3), the ratio $\sigma/r^2$ in Eq. (1) is simplified in Arthern et al. (2010) to:

$$\frac{\sigma}{r^2} = \frac{\dot{b}g}{k_g}e^{(\frac{E_g}{RT_{ave}})} \tag{4}$$

in which $\dot{b}$ is the long-term average accumulation rate ($\mathrm{kg \, m^{-2} \, s^{-1}}$) and $T_{ave}$ is the average firn temperature (K).

### 2.1.2 FDM v1.2A densification expression

Combining Eqs. (1) and (4) leads to the semi-empirical expression of Arthern et al. (2010), Eq. (B4). In FDM v1.2A, the calibration factor $\mathrm{MO}_*$ is added, leading to the dry firn densification expression:

$$\frac{\mathrm{d}\rho}{\mathrm{d}t} = \mathrm{MO}_* D \dot{b}g(\rho_i - \rho)e^{(\frac{E_c}{RT} - \frac{E_g}{RT_{ave}})} \tag{5}$$

in which $D$, which represents $k_c/k_g$, is a constant with different values above (0.03) and below (0.07) the critical density level of $\rho = 550 \ \mathrm{kg \, m^{-3}}$ to represent two distinct densification mechanisms: for $\rho < 550 \ \mathrm{kg \, m^{-3}}$, densification mainly occurs

by settling and sliding of grains, and for $\rho > 550 \ \mathrm{kg \, m^{-3}}$, it mainly occurs by deformation, recrystallization and molecular diffusion (Herron and Langway, 1980). The calibration factor $\mathrm{MO}_*$ depends on annual average accumulation and is defined separately for $\rho < 550 \ \mathrm{kg \, m^{-3}}$ ($\mathrm{MO}_{550}$) and for $550 < \rho < 830 \ \mathrm{kg \, m^{-3}}$ ($\mathrm{MO}_{830*}$). These calibration factors are based on the ratio of modelled and observed values of depths of critical density levels $\rho = 550 \ \mathrm{kg \, m^{-3}}$ ($\mathrm{z}_{550}$) and $\rho = 830 \ \mathrm{kg \, m^{-3}}$ ($\mathrm{z}_{830}$), where $\mathrm{z}_{830*} = \mathrm{z}_{830}$ - $\mathrm{z}_{550}$. $\mathrm{MO}_{550}$ and $\mathrm{MO}_{830*}$ were chosen as logarithmic and power-law functions, respectively, of

the long-term mean accumulation rate:

$$\mathrm{MO}_{550} = \alpha - \beta \ln(\dot{b}) \tag{6}$$

in which $\alpha$ and $\beta$ are fit coefficients, and

$$\mathrm{MO}_{830*} = \delta \dot{b}^{-\epsilon} + \phi \tag{7}$$

in which $\delta$, and $\epsilon$ and $\phi$ are fit coefficients.

Equations (5) to (7) use the long-term annual average accumulation rate as a proxy for overburden pressure, and Eq. (5) uses long-term annual average temperature in the grain growth part of the exponential term. To include the effect of a changing climate on firn densification previous studies with IMAU-FDM used running average accumulation and temperature over the 40 years preceding each time step (Ligtenberg et al., 2014; Munneke et al., 2014). Since there is a large spatial variation in firn age across the AIS, e.g. the firn age at the pore close off depth ranges from 20 to $\sim 3200$ years (Veldhuijsen et al., 2023a), the

firn temperature and overburden pressure change with different rates across the AIS in a warming future climate. In addition,





the change in temperature and overburden pressure also differs within the firn column. Advection and conduction transport heat vertically in the firnpack, however firn has a relatively low thermal conductivity (0.2 to 2 $\mathrm{W\,m^{-1}\,K^{-1}}$, Calonne et al. 2019). For example, Muto et al. (2011) show that firn temperature throughout a firn column differs by about 1 K due to historical temperature trends in East Antarctica of 1 to 1.5 K over 50 years. So, using a running average of the accumulation and firn
temperature is a crude approximation of the transient response of firn.

### 2.1.3 FDM v1.2AD dynamical densification expression

In this work, we aim to capture the effect of a changing climate by replacing $\dot{b}$ and $T_{ave}$ in the densification equations (Eqs. 5 to 7), while staying as close as possible to FDM v1.2A. To do so, we revert to Eq. (1) for the evolution of the snow density, including the $\mathrm{MO}_*$ term,

$$\frac{\mathrm{d}\rho}{\mathrm{d}t} = \mathrm{MO}_* k_c(\rho_i - \rho)e^{\left(\frac{E_c}{RT}\right)}\frac{\sigma}{r^2} \tag{8}$$

add $r^2$ and the age of firn layer ($A$) as prognostic variables, and calculate $\sigma$ from the weight of the overlaying firn (which includes 50 % of the weight of the layer itself). The evolution of $r^2$ for one time step includes dry-snow metamorphism (Eq. 2) and grain growth by refreezing. In case of refreezing, $r^2$ is calculated as the mass-weighted average of the solid firn grain size and refrozen liquid water grain size, under the condition that refreezing can only increase the grain size. The initial (0.03 mm) 
and refreezing (0.25 mm) grain sizes are taken from van Dalum et al. (2022). To determine $\mathrm{MO}_*$ (Eqs. 6 and 7), we replace $\dot{b}$ in those equations by the local (x,y,z) long-term mean accumulation rate ($\dot{b}_{loc}$) defined as:

$$\dot{b}_{loc} = \frac{\sigma}{A}. \tag{9}$$

In addition to capturing the effects of changes of the mean climate on the evolution of the density, these expressions also capture the effects of seasonal cycles in firn temperature and overburden pressure.
In FDM v1.2A, layer merging and splitting is constrained to the upper layer. If the upper layer thickness exceeds 0.15 m the layer is split into two equal parts, and if the layer thickness falls below 0.05 m this upper layer is merged with the layer below. Subsurface layers that become thinner than 0.05 m, due to snow compaction, are not merged. In case of snowfall, the density of freshly fallen snow is mixed with the upper layer. Since the densification rate in FDM v1.2AD depends on local overburden pressure and grain size instead of long-term annual average accumulation and temperature, this mixing of freshly fallen snow with the upper layer has a larger impact on the density evolution in FDM v1.2AD compared to FDM v1.2A. To approximate
densification of freshly fallen snow, which has a low overburden pressure and small grain size, the upper layer thickness in FDM v1.2AD is kept between 0.008 and 0.012 m. In case of snowfall in FDM v1.2AD, the age and grain size of the freshly fallen snow are mixed with the upper layer, similarly as done for the density. The splitting approach of the FDM v1.2A upper layer is then applied to the second layer in FDM v1.2AD.



## 2.2 Atmospheric forcing

IMAU-FDM is forced at the upper boundary with 3-hourly fields of snowfall, sublimation, snowdrift erosion, 10-m wind speed, surface temperature, snowmelt and rainfall from the Regional Atmospheric Climate Model RACMO2, version 2.3p2, (RACMO2.3p2). This regional climate model is used to dynamically downscale ERA5 reanalysis data (Hersbach et al., 2020) for the contemporary climate and Community Earth System Model version 2 model output (CESM2, Danabasoglu et al. 2020) for future projections to 27 km resolution.

RACMO2.3p2, driven by ERA5 reanalysis data, aims to provide an accurate description of the near-surface weather (and climate) from 1979 till the present, and has been thoroughly evaluated (RACMO2.3p2-ERA5, van Wessem et al. 2023). FDM v1.2A, forced by RACMO2.3p2-ERA5 has been extensively evaluated over the AIS in Veldhuijsen et al. (2023a) and provides firn characteristics over the AIS from 1979 till now. FDM v1.2AD has also been forced by RACMO2-ERA5 and is evaluated in this study.

van Wessem et al. (2023) used RACMO2.3p2 to dynamically downscale a historical CESM2 realization (1950-2014) and one realization of each of the low-, middle- and high-emission future (2015-2100) projection scenarios (SSP1-2.6, SSP2-4.5 and SSP5-8.5, respectively). To obtain the future evolution of Antarctic firn characteristics, FDM v1.2AD is driven by these RACMO2.3p2-CESM2 realizations. We also run FDM v1.2A for the SSP5-8.5 scenario and compare this to FDM v1.2AD SSP5-8.5. CESM2 simulates the coupled interactions between the atmosphere-ocean-land components of the climate system on the global scale at 1 degree horizontal resolution (Danabasoglu et al., 2020), and has been thoroughly evaluated over the AIS (Gorte et al., 2020; Dunmire et al., 2022). The model has a relatively detailed representation of polar processes and is among the best CMIP6 models in representing the present Antarctic SMB (Gorte et al., 2020). The projected Antarctic warming in SSP5-8.5 (+7.7 °C) is stronger than the mean CMIP6 warming (+5 °C), which enables us to assess the AIS firn layer response to strong warming.

RACMO2.3p2-CESM2 time series of mean annual surface temperature, accumulation and surface melt over the AIS are shown in Fig. 1. Compared to RACMO2.3p2-ERA5 for the overlapping period (1979-2014) we find a temperature bias of -1.2 °C over the AIS, an accumulation bias of -7.6 $\mathrm{mm\,yr^{-1}}$ (-4 %) and a snowmelt bias of -1.0 mm (-11 %). These biases can be explained by the cold bias in CESM2 (Dunmire et al., 2022). RACMO2.3p2-CESM2 is also used by van Wessem et al. (2023) in their assessment of future meltwater ponding and compares well to meltwater lake volume observations of the Sentinel-2 satellite. Maps of the differences in mean annual surface temperature, accumulation and surface melt in the period 1979-2014 between RACMO2.3p2-CESM2 and RACMO2.3p2-ERA5 are shown in Fig. S1, and further discussed in Section 3.2.

For all scenarios we find that accumulation increases with increasing atmospheric temperatures. Under SSP5-8.5, the surface temperature increases by 6.7 °C in 2090-2100 compared to 2005-2014, average accumulation increases to 270 $\mathrm{mm\,yr^{-1}}$ (+46 %), and average surface melt increases to 74 $\mathrm{mm\,yr^{-1}}$ (+924 %). In SSP1-2.6 and SSP2-4.5 the surface temperature increases by 2.2 and 3.1°C, the accumulation to 209 $\mathrm{mm\,yr^{-1}}$ (+14 %) and 213 $\mathrm{mm\,yr^{-1}}$ (+16 %) and the surface melt increases to 21 $\mathrm{mm\,yr^{-1}}$ (+195 %) and 27 $\mathrm{mm\,yr^{-1}}$ (+272 %).



Under SSP5-8.5, the surface temperature over ice shelves increases by 6.9 °C in 2090-2100 compared to 2005-2014, average accumulation increases to 374 mm yr$^{-1}$ (+15 %), and surface melt increases to 292 mm yr$^{-1}$ (+777 %). In SSP1-2.6 and SSP2-4.5 the surface temperature over ice shelves increases by 2.2 and 3.1 °C, the accumulation to 353 mm yr$^{-1}$ (+8 %) and 361 (+11 %) and the surface melt increases to 91 mm yr$^{-1}$ (+172 %) and 112 mm yr$^{-1}$ (+235 %). Rainfall over the AIS increases from 0.4 mm yr$^{-1}$ (2005-2014) to 1.2, 1.5 and 7.5 (+195, +264, +1763 %) mm yr$^{-1}$ in 2090-2100 for SSP1-2.6, SSP2-4.5, SSP5-8.5, respectively.

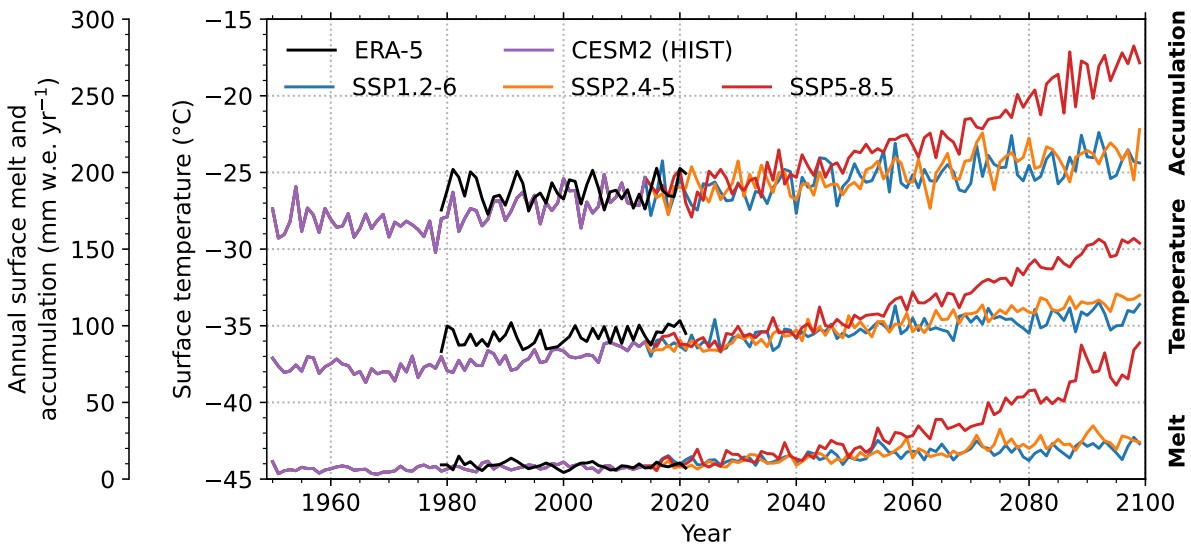

**Figure 1.** Time series of mean annual surface melt, accumulation and surface temperature over the AIS from RACMO2.3p2 forced by ERA5 (1979-2021), historical CESM2 (1950-2014) and future CESM2 scenarios SSP1-2.6 (2015-2100), SSP2-4.5 (2015-2100) and SSP5-8.5 (2015-2100).

## 2.3 Experimental setup

IMAU-FDM versions used in this study differ by the calculation of dry-snow densification and model tuning. Abbreviations and characteristics of these versions are listed in Table 1. For all IMAU-FDM simulations driven by RACMO2.3p2-ERA5, an initial firn layer is obtained by looping over the forcing of the 1979-2020 reference period, since no significant AIS-wide long-term trends in surface climate have been detected during that period. In contrast to RACMO2.3p2-ERA5, RACMO2.3p2-CESM2 does exhibits AIS-wide long-term trends in the modelled historical (1950-2014) surface climate (Fig. 1). We therefore used the 1950-1999 period to initialise IMAU-FDM simulations driven by RACMO2.3p2-CESM2.





**Table 1.** Abbreviations and characteristics of IMAU-FDM versions used in this study.

| Abbreviation | Forcing | Dry-snow densification |
|---|---|---|
| FDM v1.2A-E[a] | RACMO2.3p2, ERA5 | Arthern et al. (2010) (Eq. 5) |
| FDM v1.2A-C | RACMO2.3p2, CESM2 | Arthern et al. (2010)(Eq. 5) |
| FDM v1.2AD-E | RACMO2.3p2, ERA5 | Dynamical version (Eq. 8) |
| FDM v1.2AD-C | RACMO2.3p2, CESM2 | Dynamical version (Eq. 8) |

[a] Veldhuijsen et al. (2023a)

## 2.4 In situ measurements

To calibrate and evaluate the firn model we compare the historical part (1950-2014) of the simulation to in situ firn core density measurements (Fig. 2). We used 112 density profiles across the AIS, by combining multiple published datasets (van den Broeke, 2008; Schwanck et al., 2016; Bréant et al., 2017; Fernandoy et al., 2010; Montgomery et al., 2018; Fourteau et al., 2019; Olmi 195 et al., 2021; Winstrup et al., 2019). Detailed information about the dataset is presented in Veldhuijsen et al. (2023a).

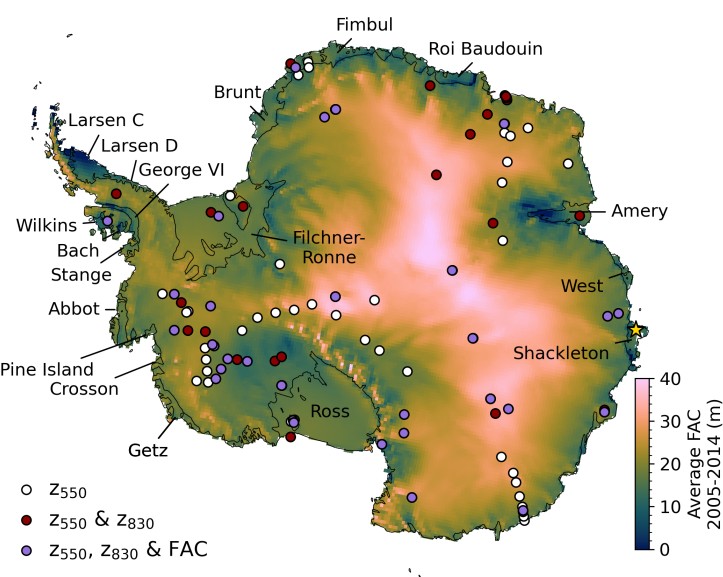

**Figure 2.** Map of Antarctica showing average firn air content (FAC) for the period 2005-2014 from FDM v1.2AD-C. The white circles indicate locations of in situ observations of depths of critical density level $\rho = 550$ kg m$^{-3}$ ($z_{550}$), the red circles of depths of critical density levels $\rho = 550$ kg m$^{-3}$ ($z_{550}$) and $\rho = 830$ kg m$^{-3}$ ($z_{830}$), the purple circles of depths of the critical density levels $\rho = 550$ kg m$^{-3}$ ($z_{550}$) and $\rho = 830$ kg m$^{-3}$ ($z_{830}$), and of firn air content (FAC). The yellow star indicates the location of Fig. 3b. The names indicate the ice shelves referred to in the text.





## 2.5 Accessible firn air content

Meltwater refreezing can form ice slabs, which can impede meltwater percolation to deeper firn, thereby reducing the firn layer's meltwater buffering capacity (MacFerrin et al., 2019; Culberg et al., 2021). Hence, in this study, in addition to the FAC (henceforth referred to as total FAC), we also calculate the accessible FAC, as defined below. To do so, we use ice layer thickness as a measure of their permeability. It is important to note that the calculation of accessible FAC is a post-processing step, and ice layers within the firn model itself are permeable for vertical liquid water movement.

Figure 3a shows relationships between ice layers thickness and permeability (the permeability factor) from several observational studies. A small-scale field experiment conducted in Greenland shows that ice layers of 0.12 m can still be completely permeable for liquid water (Samimi et al., 2020). Samimi et al. (2021) assume that ice layers thicker than 0.5 m act as impermeable barriers and prescribe a non-linear decrease in permeability between 0.1 and 0.5 m. On the other hand, in another small-scale field experiment in Greenland, ice lenses of only 3-5 cm have also been found to be partly impermeable (Clerx et al., 2022). These results stress that processes of meltwater percolation and refreezing occur at a small scale. Firn is therefore spatially heterogeneous (e.g., Samimi et al., 2020; Vandecrux et al., 2019), and on a larger horizontal scale, such as that of a model grid cell (27 km), ice layers can be discontinuous, allowing meltwater to still percolate through. We assume that lateral connectivity of ice layers increases with ice thickness, and for ice layers to be impermeable on a large scale, such as in IMAU-FDM simulations, this requires at least a larger thickness than based on small-scale field experiments.

Radar data and firn cores show that horizontally continuous > 1 m thick ice slabs develop on top of refrozen ice layers after extreme melt years. A radar study on Devon Ice Cap, Canada, revealed that an initially widespread ice layer that formed during an extreme melt year, thickened by between 0.5 and 4.5 m over the subsequent 5 years (Gascon et al., 2013). Similarly, over time thickening ice layers of 1-2 m are formed in Greenland Culberg et al. (2021) following an extreme melt year. While these large-scale radar observations do not give an exact relation between thickness and permeability, they do give an indication that ice layers thicker than 0.5 m are at least partly impermeable on a larger scale. Based on these observations, we propose a relationship between ice layer thickness ($z$) and permeability factor ($Pf$) (Fig. 3a), using a sigmoid function:

in which $a$ and $b$ are tuning coefficients, for which we propose $a = 1.130$ and $b = 3.245$.

$$Pf = \frac{1 + b + (az)}{b + e^{(az)}} \tag{10}$$

Since exact observations that evaluate the permeability as a function of ice layer thickness at the regional scale are lacking, the relationship in Eq. (10) must be regarded as a rough estimate. In addition, the location of the mentioned observations have a stronger surface slope (> 0.4 °) (Yi et al., 2005) than most Antarctic ice shelves (< 0.15 °) (Slater et al., 2018). The low surface slopes on ice shelves results in less lateral flow on top of refreezing layers which may impact the permeability. Considering these uncertainties, we test the sensitivity of our results to a range of possible relations, indicated by the envelope of the black shaded surfaces in Fig. 3a, in which we adjusted the values of $a$ (1.119 and 1.250) and $b$ (13.490 and 0.0594). In addition, we also assess the total FAC, which represents full permeability.

To calculate the accessible FAC of a layer, its FAC is multiplied with the permeability factors of all overlying ice layers. E.g., if two distinct ice layers with $Pf = 0.5$ are overlying a firn layer, the FAC of that layer is multiplied twice by 0.5 to yield the





accessible FAC of that layer. The sum of accessible FAC of the individual layers equals the accessible FAC. An ice layer can range from a single to numerous model layers. Impermeable ice layers are usually defined as having a density $> 830 \text{ kg m}^{-3}$ (the pore close-off density). Here, we use a threshold of $> 900 \text{ kg m}^{-3}$, which corresponds to the density of near-surface refreezing ice layers. This choice limits the impact on the accessible FAC of changes in high-density non-refreezing layers in the deep firn. Figure 3b shows the impact of ice layer formation on accessible FAC for an example location in a high-end warming scenario. As can be seen, ice layer formation from 2060 onwards depletes the accessible FAC compared to the total FAC. Henceforth, we refer to (a set of) ice layers that have a substantial impact on the accessible FAC as ice slabs.

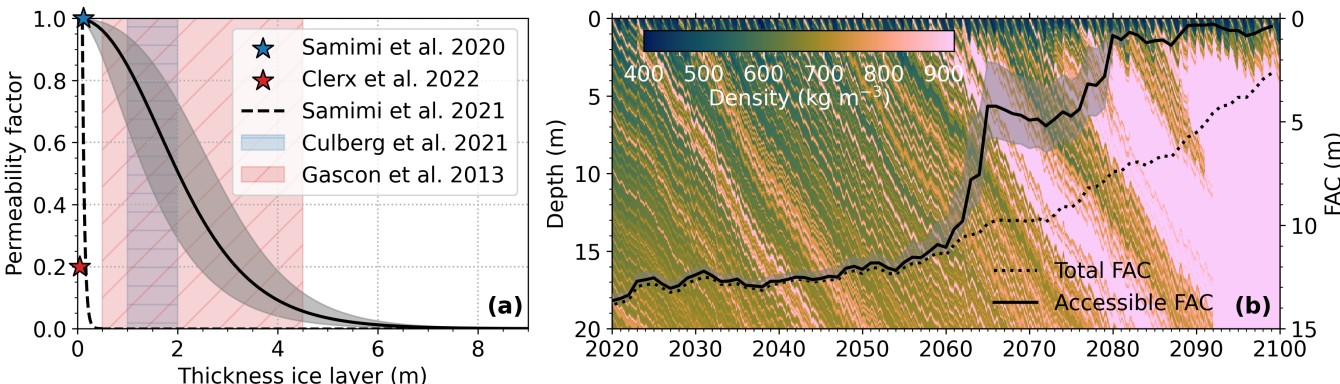

**Figure 3.** Relations between the ice layer thickness and permeability factor, including results from in situ observations (stars), a previous estimate (dotted line), observed ranges of reduced permeability from radar observations (colored shaded surfaces) and a relationship proposed here (solid line) including tested sensitivity ranges (black shaded surface). **(b)** Example plot of simulated density including total FAC and accessible FAC (calculated using the proposed relationship from panel **(a)** simulated with FDM v1.2AD-C for SSP5-8.5. The location is at Shackleton ice shelf and is indicated by the yellow star in Fig. 2.

## 3 Calibration and model performance

### 3.1 Calibration

The densification equation of FDM v1.2A-E has been calibrated in Veldhuijsen et al. (2023a). To calibrate the dry-snow densification rate of FDM v1.2AD, we first performed simulations of FDM v1.2AD-E and FDM v1.2AD-C for locations with firn density observations (Fig. 2), without MO corrections, i.e. in which the MO values are equal to 1. The resulting MO fits and statistics are shown in Fig. 4. FDM v1.2AD-E yields similar $R^2$ (0.39 and 0.86) compared to FDM v1.2A-E (0.37 and 0.88). The $MO_{830*}$ fits of these models are similar, while the $MO_{550}$ fit of FDM v1.2AD-E is steeper and higher, which is due to a relatively low ratio of overburden pressure and grain size in the upper firn $(< z_{550})$. In addition, the CESM2 forcing again alters the $MO_{550}$ and $MO_{830*}$ fits, due differences in modelled accumulation rates and surface temperature (see Section 2.2), but the fit quality remains similar ($R^2$ is 0.43 and 0.86 for $MO_{550}$ and $MO_{830*}$, respectively).



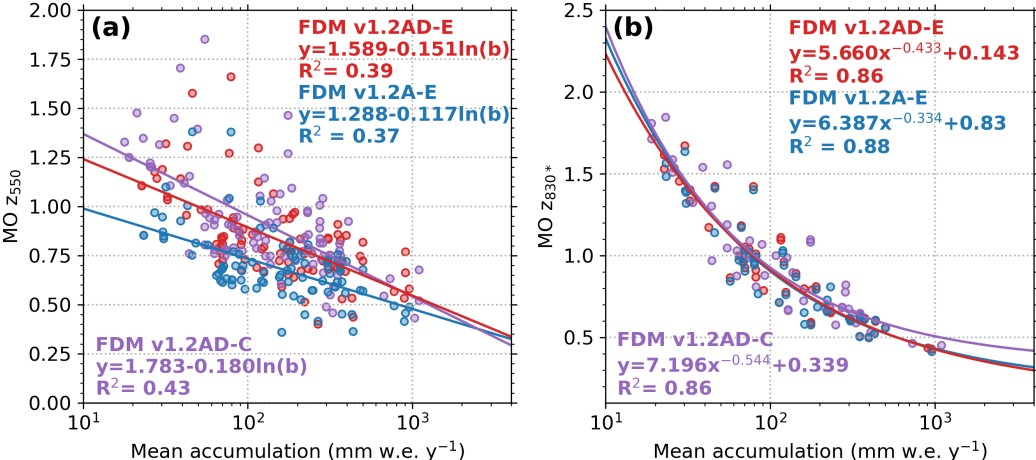

**Figure 4.** MO ratios and fits for FDM v1.2AD-E, FDM v1.2A-E and FDM v1.2A-C for **(a)** $z_{550}$ and **(b)** $z_{830*}$ as a function of the annual average accumulation. 2.

## 3.2 Model performance

We evaluate the performance of FDM v1.2AD by comparing it with FDM v1.2A and in situ observations. For the latter, we compare simulated to observed depths of the critical density levels $\rho = 550 \, \text{kg m}^{-3}$ ($z_{550}$) and $\rho = 830 \, \text{kg m}^{-3}$ ($z_{830}$) and FAC

(Figs. 5a,b and c). FDM v1.2AD-E yields similar root mean square error (RMSE) and bias compared to FDM v1.2A-E for $z_{550}, z_{830*}$ and FAC, which indicates a similar performance of the updated firn model for the current climate. FDM v1.2AD-C yields slightly higher RMSE than FDM v1.2AD-E for $z_{550}, z_{830*}$ and FAC, indicating that the CESM2 forcing, which is not constrained by observations like ERA5, results in a slightly poorer performance.

The absolute FAC difference between FDM v1.2AD-E and FDM v1.2A-E over the period 1979-2020 is on average only 1

%, with some spatial variations (Fig. 5d). Over the grounded ice, the differences are even smaller (0.6 %), whereas they are higher in regions with high MOA, such as on Wilkins and Larsen C ice shelves, and around the grounding lines of e.g. Roi Baudouin and Amery ice shelves. The reason is that densification is enhanced in regions with high MOA, resulting in lower FAC values (Fig. 5e). Figure 5f shows the difference in FAC over the AIS between FDM v1.2AD-C and FDM v1.2A-E for the period 1979-2014. The average absolute difference is 5.2 %, which is caused by a combination of the updated dry-snow

densification expression (Fig. 5d) and different climatic forcing. Figure S1 shows the difference in mean annual accumulation, surface temperature and surface melt between RACMO2.3p2-CESM2 and RACMO2.3p2-ERA5 for the period 1979-2014 over the AIS. Accumulation and temperature in RACMO2.3p2-CESM2 are in general lower, except in Dronning Maud Land and Enderby Land. Temperatures are most notably lower on the Ross ice shelf and in West Antarctica. Surface melt is in general lower, except in Dronning Maud Land and on Larsen C ice shelf. The difference in FAC evolution in response to future

warming scenarios between FDM v1.2AD and FDM v1.2A is discussed in Section 5.1.





**Figure 5.** Simulated against observed **(a)** $z_{550}$, **(b)** $z_{830*}$ and **(c)** firn air content (FAC) for FDM v1.2AD-E, FDM v1.2A-E and FDM v1.2AD-C. **(d)** Difference in average FAC between FDM v1.2AD-E and FDM v1.2A-E for the period 1979-2020. **(f)** FAC distribution by melt-over-accumulation ratio (MOA) bins of 0.1 for FDM v1.2AD-E and FDM v1.2A-E averaged for the period 1979-2020. **(e)** Difference in average FAC between FDM v1.2AD-C and FDM v1.2A-E for the period 1979-2014.

## 4 Results

### 4.1 Total and accessible firn air content in AD 2100

Using the updated firn model FDM v1.2AD, we project the firn evolution over the AIS for the period 1950-2100 forced by RACMO2.3p2-CESM2 for climate scenarios SSP1-2.6, SSP2-4.5 and SSP5-8.5. Figures 6a,b and c show the relative total

FAC change by 2090-2100 compared to 2005-2014. Over the grounded ice some regions experience an increase and others a decrease in FAC. On average, FAC over the grounded ice decreases by 1.0 %, 1.5 % and 2.4 % for scenarios SSP1-2.6, SSP2-4.5 and SSP5-8.5, respectively. This implies that according to IMAU-FDM, the projected warming, leading to lower FAC, is almost balanced by enhanced precipitation, leading to higher FAC. This is different for most ice shelves, where we



find that FAC decreases by 15 %, 19 % and 42 % for scenarios SSP1-2.6, SSP2-4.5 and SSP5-8.5, respectively. For SSP1-2.6
and SSP2-4.5 we find a substantial FAC decrease on Larsen C (70 % and 75 %), Wilkins (59 and 59 %), Roi Baudouin (67
and 81 %), George VI (46 and 56 %) and Bach (59 and 66 %) ice shelves. For SSP5-8.5, in addition we find a substantial FAC
decrease for ice shelves in Dronning Maud Land, such as Fimbul ice shelf (-91 %), and elsewhere for Abbot (-76 %), Pine
Island (-74 %), West (-83 %) and Shackleton (-76 %) ice shelves. FAC decrease on Ross and Filchner-Ronne ice shelves is
more modest (-21 %) and (-19 %).

Figures 6d, e and f show the relative change in accessible FAC by 2090-2100 compared to 2005-2014. Ice layer formation
by refreezing mainly occurs over the ice shelves, and is limited over the grounded ice sheet, except for the region east of the
Ross ice shelf under SSP5-8.5. For SSP1-2.6 the decrease of accessible FAC is accelerated compared to the total FAC on the
Roi Baudouin ice shelf (-79 vs -67 %), but on average the difference over ice shelves is limited (-18 vs -15 %). For SSP2-4.5
the depletion is significantly accelerated on e.g. Fimbul (-56 vs -44 %), Amery (-44 vs –30 %) and Larsen D (-67 % vs -49 %)
ice shelves. On average, the difference over ice shelves under this scenario is limited as well (-23 vs -19 %). For SSP5-8.5 the
accessible FAC depletion due to ice layer formation is significantly accelerated compared to total FAC depletion on most ice
shelves (on average -53 % vs -42 %), such as on Amery (-91 vs -53 %), Shackleton (-94 vs 76 %), Brunt (-93 vs -74 %) and
Filchner-Ronne (-19 vs -33 %).

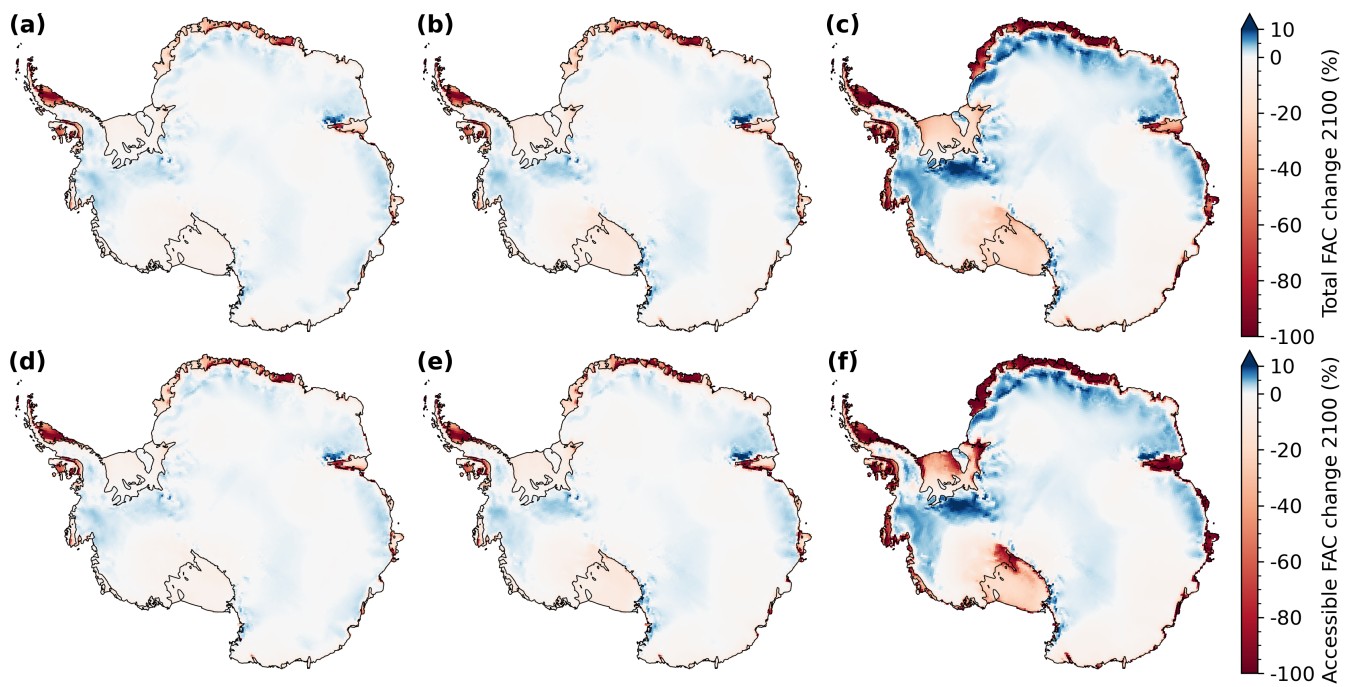

**Figure 6.** Relative change in total firn air content (FAC) (top row) and accessible FAC (bottom row) by 2090-2100 compared to 2005-2014
for **(a,d)** SSP1-2.6, **(b,e)** SSP2-4.5 and **(c,f)** SSP5-8.5. Please note the different scales for decreasing (red) and increasing (blue) FAC.





## 4.2 Climatic drivers of changes in firn air content

In this section, we compare changes in FAC under SSP5-8.5 to various climate variables. Due to low temperatures in the interior ice sheet, 76 % of the AIS does not experience melt by the end of the century even in this strong warming scenario. Here, changes in FAC are solely driven by increasing temperature and changing accumulation rates. For 42 % of those melt-free locations, FAC decreases by 2090-2100, and for 58 %, it increases. In Fig. 7a we show the relative change in total FAC by 2090-2100 compared to 2005-2015 for these non-melt locations as a function of the temperature and accumulation change. For

4-6 °C warming an increase of at least 30 % accumulation is needed to compensate for the increased densification, whereas for > 8 °C warming, an increase of at least 70 % accumulation is required. Because of these compensating mechanisms, for most of the AIS the FAC remains relatively stable.

When we compare the change in FAC of the entire AIS to current climate conditions (1950-2014, Fig. 7b), we observe the largest relative decrease in FAC (-49 %) in currently warm regions (> -22 °C) receiving less than 1000 mm accumulation

annually. For high-accumulation regions (> 1,000 $\mathrm{mm\,yr^{-1}}$), such as parts of Getz ice shelf and the northwestern part of the Antarctic Peninsula, the decrease in FAC is notably smaller (-22 %). FAC also slightly decreases (-14 %) in colder regions (-34 and -24 °C) with low accumulation (< 100 $\mathrm{mm\,yr^{-1}}$). Large differences between total FAC and accessible FAC mainly occur in current intermediate warm (-28 to -16 °C) and relatively dry (< 500 $\mathrm{mm\,yr^{-1}}$) locations (0.85 m) (Fig. 7c), and most prominently in regions with temperatures between -22 and -18 °C and accumulation rates between 100 and 400 $\mathrm{mm\,yr^{-1}}$ (1.6

m). For colder initial conditions, the projected melt is too weak to allow for ice lens formation, while for warmer locations, the firn layer is projected to completely disappear in 2090-2100. In the wettest locations (> 600 $\mathrm{mm\,yr^{-1}}$), the average differences are smaller (0.55 m). The patterns in Fig. 7 are in general similar for SSP1-2.6 and SSP2-4.5, albeit with smaller magnitudes (Fig. S2).

## 4.3 Temporal evolution of total FAC, accessible FAC and runoff over ice shelves

Time series of total FAC, accessible FAC and runoff extent for 12 major ice shelves under SSP2-4.5 and SSP5-8.5 are shown in Fig. 8 (See Fig. S3 for SSP1-2.6). From 2020 onwards there is a gradual decrease in FAC on all ice shelves for all scenarios. Enhanced FAC depletion on Antarctic Peninsula (Larsen C, Wilkins and George VI) ice shelves and Roi Baudouin ice shelf occurs around 2030 for all scenarios. Enhanced FAC depletion on Fimbul, Abbot, Pine Island and Shackleton ice shelves occurs around 2050-2060 for SSP5-8.5. For SSP1-2.6 and SSP2-4.5 enhanced FAC depletion occurs around 2050 for Fimbul

ice shelf. For SSP2-4.5 less than 6 m FAC is left on the Antarctic Peninsula ice shelves and Roi Baudouin ice shelf by 2100. For SSP5-8.5 FAC decreases by more than 4 m on all ice shelves resulting in less than 3 m FAC left on Wilkins, Larsen C, George VI, Fimbul, Abbot and Shackleton ice shelves by 2100.

Differences between the accessible and total FAC mainly occur under SSP5-8.5, and are most pronounced on Shackleton, Fimbul, Pine Island, Roi Baudouin, Amery and Filchner-Ronne ice shelves, where depletion is accelerated by up to 20 years

and up to 5 m. These ice shelves have in common that they are currently relatively dry (< 500 $\mathrm{mm\,yr^{-1}}$). On most ice shelves, the difference increases gradually. However, at Fimbul and Shackleton ice shelves, ice layer formation around 2060 results in




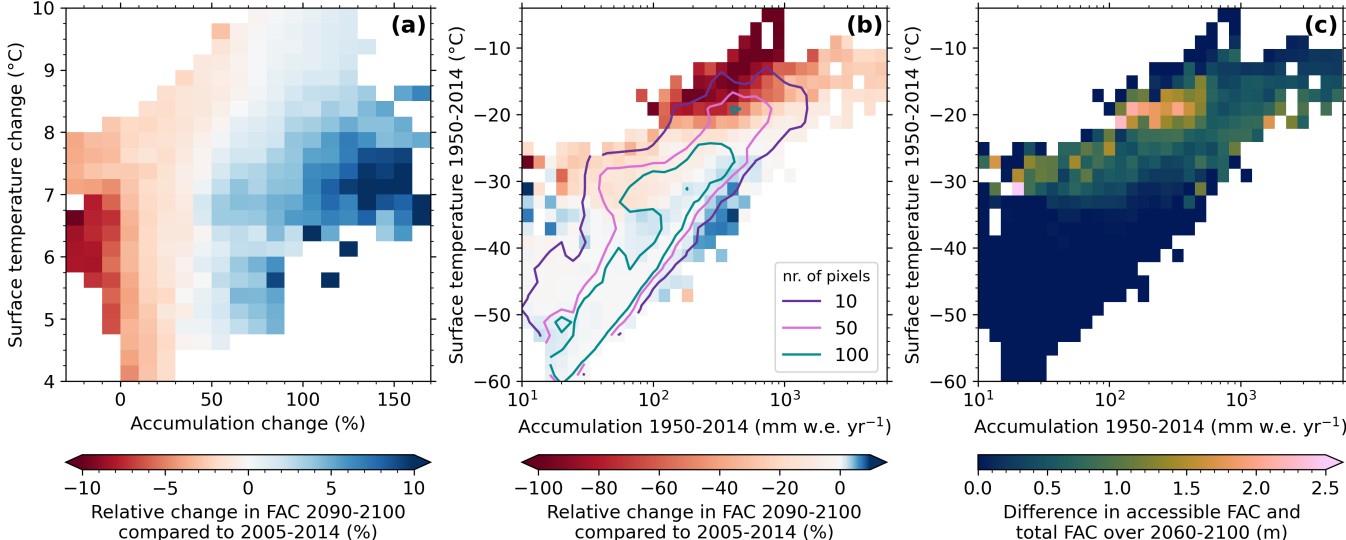

**Figure 7.** Relative change in total FAC by 2090-2100 for SSP5-8.5 compared to 2005-2014 **(a)** as a function of temperature change and accumulation change by 2090-2100 compared to 1950-2014 for locations that do not experience melt by the end of the century in SSP5-8.5 and **(b)** as a function of annual average accumulation and temperature (1950-2014) for the entire AIS. **(c)** Difference between accessible firn air content and total firn air content in 2060-2100 for the entire AIS for SSP5-8.5 as a function of annual average accumulation and temperature (1950-2014). Contour lines in **(b)** indicate the number of pixels per bin. Please note the different scales for decreasing (red) and increasing (blue) FAC in panel **(b)**.

a quick depletion of more than 5 m (> 50 %) accessible FAC in 5 years. This coincides with high melt rates (+25 % higher on Fimbul and +38 % higher on Shackleton compared to the previous and following 5 years). On the other hand, on the currently relatively warm (> -19 °C) and wet (> 600 mm yr$^{-1}$) Wilkins, Getz, Abbot and George VI ice shelves there remains little
difference between accessible and total FAC, which corresponds with Fig. 7c. For SSP2-4.5 there is generally little difference between total FAC and accessible FAC, except for Roi Baudouin, Fimbul and Amery ice shelves (>1 m reduction in accessible FAC compared to total FAC after 2070).

    The runoff extent is the arial fraction of the ice shelf where runoff is generated in a given year, i.e. where the firn layer has no or only limited meltwater storage capacity left. Since ice layers are fully permeable for meltwater percolation in IMAU-FDM,
the runoff time series are closely related to the total FAC time series, with some differences (Figs. 8e,f). For Wilkins, Abbot and George VI ice shelves the runoff extent is large compared to the FAC depletion (>50 % runoff extent with 5 m FAC), in contrast to Shackleton and Fimbul ice shelves, where the runoff extents are only 44 % and 40 %, respectively, with 2.5 m FAC. The reason for the high runoff extent on Wilkins, Abbot and George VI is the combination of high accumulation and melt rates. The high melt allows for saturation of a thick firn layer, which is maintained by high accumulation. In addition, runoff here
also occurs year-round from firn aquifers, which are perennial subsurface bodies of liquid water, that become more ubiquitous



in a warmer Antarctica (Bell et al., 2018). On drier ice shelves, the amount of melt is apparently not enough to saturate the firn column, even though FAC is low.

For Wilkins ice shelf we see a quick increase from 0 to > 90 % runoff extent for both scenarios, which indicates a limited spatial variation in FAC depletion for grid points across the ice shelf. On the other hand, Larsen C, a larger ice shelf with large 340 latitudinal extent and climate gradients, shows a gradual increase in runoff extent, revealing higher spatial variability in its response to warming. On average, 6 % and 25 % of the entire Antarctic ice shelf area experiences runoff in 2090-2100 under SSP2-4.5 and SSP5-8.5, respectively, indicated by the black line.

**Figure 8.** Time series of **(a,b)** total firn air content (FAC), **(c,d)** accessible FAC and **(e,f)** runoff extent of 12 ice shelves simulated with FDM v1.2AD-C for **(a,c,e)** SSP2-4.5 and **(b,d,f)** SSP5-8.5 for the period 2015-2100. The shaded areas indicate the sensitivity to the relation between ice layer thickness and permeability factor shown in Fig. 3a.



### 4.4 Ice layer formation and its climatic drivers

In the previous sections, we showed that ice layer formation on some ice shelves causes enhanced depletion of accessible FAC. Figures 9a and b respectively show the maximum absolute difference and the corresponding relative difference between total FAC and accessible FAC that occurs over the period 1950-2100 for SSP5-8.5. The differences are highest on ice shelves in Dronning Maud Land and on the Amery, West, Shackleton, Ross and Filchner-Ronne ice shelves. In contrast, for Ross the maximum difference is found near the grounding line, and for Filchner-Ronne near the seaward edge. The differences are lowest in the Bellingshausen Sea region, on ice shelves as Wilkins, George VI and Stange and Abbot, and on the Getz and Crosson ice shelves (< 3.9 m and < 47 %), which are among the warmest and wettest ice shelves of the AIS (> -19 °C and > 600 mm yr$^{-1}$), which is in line with Fig. 7c.

In Figs. 9c to h we zoom in on FAC and accessible FAC time series for selected locations plotted together with accumulation, melt and associated MOA. At Amery, Shackleton, Filchner-Ronne and Larsen C ice shelves, we see that extreme melt seasons can cause a persistent reduction in accessible FAC. The 5-year running average MOA values for which this occurs on Amery, Shackleton and Filchner-Ronne are between 0.59-0.71. On Larsen C, where melt is more constant over the years, this occurs for a MOA of about 1.04. On the wetter Wilkins location, ice layer formation is limited, even though the firn layer becomes completely depleted and a MOA of 1 is exceeded. In addition, on the wetter Getz ice shelf location, ice layer formation is also limited as a MOA of 0.6 has not been reached, even though > 50 % of the FAC has become depleted. The general pattern of these figures is that melt-water blocking ice lenses are primarily formed in drier locations with significant interannual variability in melt.

### 4.5 Differences in projections from FDM v1.2AD and FDM v1.2A

The change to a dynamical densification expression changes the temporal evolution of the firn. In this section and in Fig. 10, we quantify this effect by comparing the FAC by 2080-2100 under SSP5-8.5 simulated by FDM v1.2A and FDM v1.2AD. As described in Section 2.1, both models include the effect that warmer snow densifies faster. However, in FDM v1.2A, the compensating effects of enhanced grain growth, which makes snow stiffer, and enhanced accumulation, which increases the overburden pressure, are parameterized using 40 year running averages of temperature and accumulation.

High accumulation rates result in firn mostly younger than 40 to 100 years, and in such regions FDM v1.2A provides lower 2080-2100 FAC estimates than FDM v1.2AD. This indicates that the effect of grain size growth, which slows down densification, is underestimated in FDM v1.2A in a transient climate for these locations. Figure 10b shows that these young firn locations are mostly found along the coasts of the Antarctic Peninsula and West Antarctica, such as Wilkins, Getz and George VI ice shelves. Average end of century (2080-2090) FAC estimates of FDM v1.2AD are 3.8 and 1.1 m higher compared to FDM v1.2A in regions with a maximum firn age lower than 40 and 100 years, respectively.

For locations with older firn (> 100 years), the differences between FDM v1.2A and FDM v1.2AD are smaller. The root mean square difference (RMSD) is only 0.22 m and the bias is -0.07 m. This does not prove that the parameterizations of transient behavior are correct, but at least their errors balance out largely. Clear spatial patterns are visible in Fig. 10b.



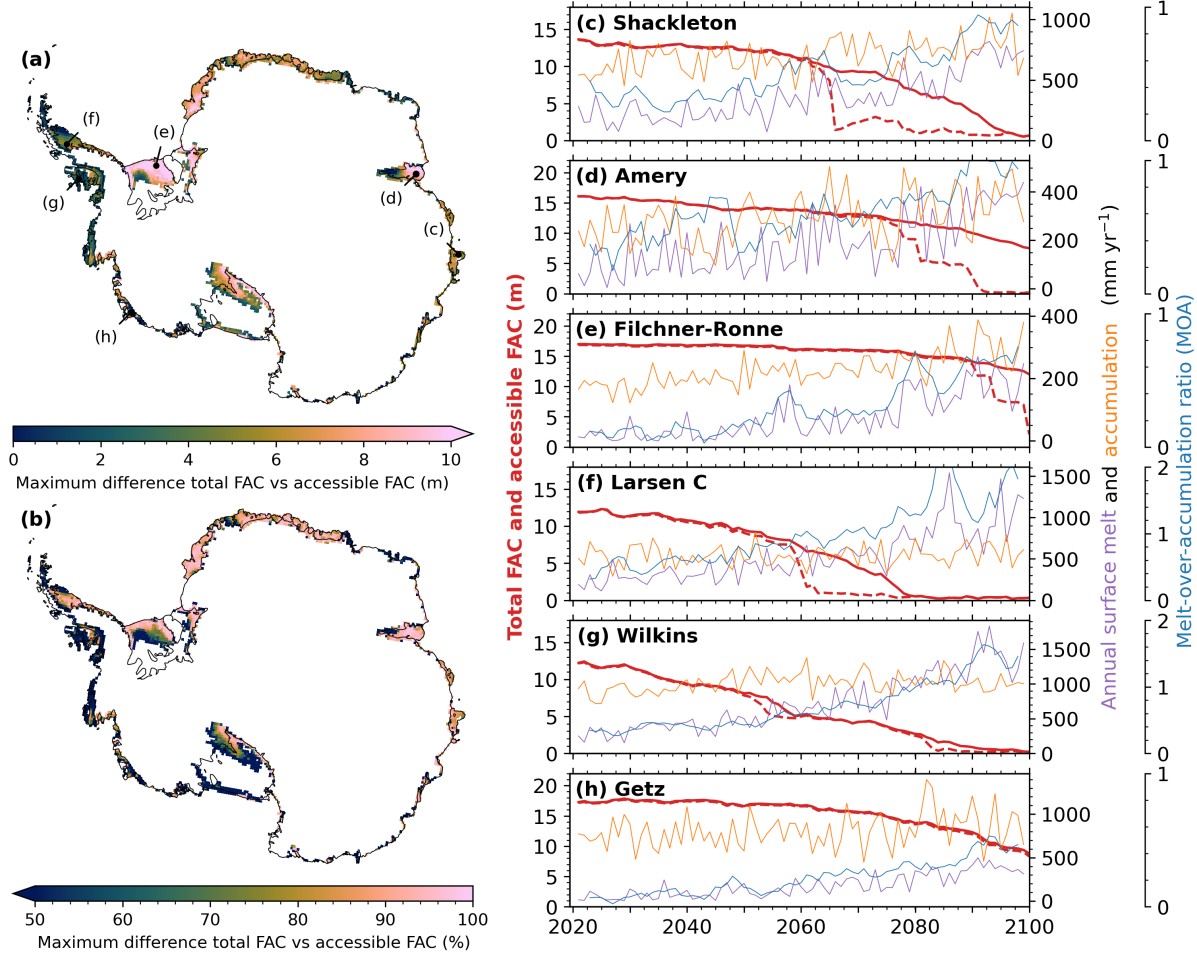

**Figure 9. (a,b)** Maximum difference between firn air content (FAC) and accessible FAC in FDM v1.2A (CESM2 SSP5-8.5) simulation (1950-2100) shown for locations with at least 25 % accessible FAC depletion in 2090-2100 compared to 2005-2014. (c-h) Time series of total FAC (solid red line), accessible FAC (dashed red line), annual surface melt, annual accumulation and 5-year running average melt-over-accumulation ratio (MOA) ratio for individual grid points on **(c)** Shackleton, **(d)** Amery, **(e)** Filchner-Ronne, **(f)** Larsen C, **(g)** Wilkins and **(h)** Getz ice shelves. The locations of the grid points are indicated in panel **(a)**.

At the currently relatively dry Ross and Filchner-Ronne ice shelves (105 and 170 $\mathrm{mm\,yr^{-1}}$ annual average accumulation, respectively) FDM v1.2AD results in a quicker FAC depletion (-0.45 m and -0.41 m, respectively) compared to FDM v1.2A, due to a dominating effect of the increase in temperature (+8.1 K and + 8.4 K, respectively) compared to accumulation (+10 % and +61 %, respectively) by 2080-2100 compared to 1950-2014.



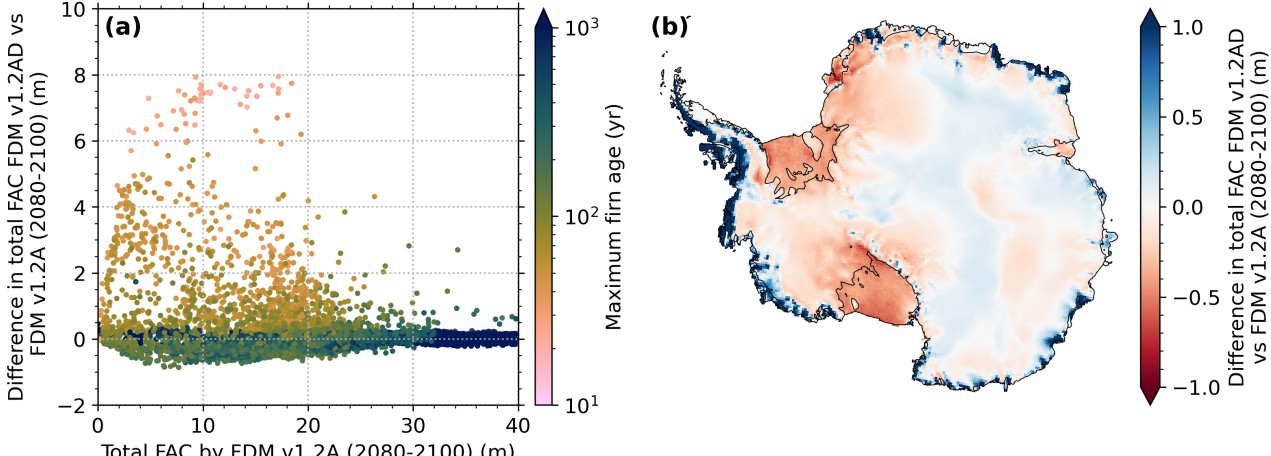

**Figure 10. (a)** Average total firn air content (FAC) difference between FDM v1.2AD-C and FDM v1.2A-C against average FAC for FDM v1.2A-C for the period 2080-2100. The color indicates the maximum firn age (taken as the age at the critical density level $\rho = 830 \ \mathrm{kg \ m^{-3}}$) from FDM v1.2A-C (1950-2014). **(b)** Average total FAC difference between FDM v1.2AD-C and FDM v1.2A-C for the period 2080-2100.

## 5   Discussion

### 5.1   Firn air depletion

We use a firn model with an updated dynamical densification parameterization to simulate FAC over the AIS under future warming scenarios, and we include the impact of ice slabs on the accessible FAC. The updated model, FDM v1.2AD, yields lower FAC compared to FDM v1.2A for regions with high MOA (Fig. 5f), which aligns with theoretical considerations that a MOA exceeding 0.7 initiates depletion of the pore space (Pfeffer et al., 1991; van Wessem et al., 2023). Apart from that, the simulated densities with FDM v1.2A and FDM v1.2AD over the current climate are similar. When using FDM v1.2AD in a warming transient climate (here tested for SSP5-8.5), the Ross and Ronne-Filcher ice shelves are more vulnerable to firn air depletion than based on previous FDM modelling studies (Munneke et al., 2014), on the other hand, high-accumulation ice shelves, such as Getz, Wilkins, George VI are less vulnerable (Fig. 10).

The high vulnerability of the Antarctic Peninsula ice shelves and Roi Baudouin ice shelf to total FAC depletion (> 50 % decrease in FAC for all scenarios) aligns with previous studies (Gilbert and Kittel, 2021; Munneke et al., 2014; van Wessem et al., 2023). According to our results, enhanced FAC depletion on these ice shelves is expected to start around 2030 for all scenarios (Figs. 8, S3). For SSP5-8.5, we also find a substantial FAC depletion (> 76 % decrease) for ice shelves in Dronning Maud Land, such as Fimbul, and elsewhere for Abbot, Pine Island, West and Shackleton ice shelves. Enhanced FAC depletion on these ice shelves is expected to occur around 2050-2060 for SSP5-8.5.

By including the effect of reduced permeability of ice slabs, our results demonstrate enhanced FAC depletion under SSP5-8.5 on ice shelves with current accumulation rates of $< 500 \ \mathrm{mm \ yr^{-1}}$ and mean annual temperatures of $< -16 \ °C$ and significant



interannual variability in melt. Such ice shelves are mainly situated in East Antarctica, but also include the Ross, Filchner-Ronne, Larsen C and Pine Island ice shelves. This enhanced depletion starts around 2060-2070 for Shackleton, Fimbul and

Pine Island shelves ($\sim$ 4 °C AIS-wide average warming), and around 2080 for the colder Ross and Filchner-Ronne ($\sim$ 6 °C AIS-wide average warming). For SSP1-2.6 and SSP2-4.5 the accelerated FAC depletion by ice slabs is limited to Roi Baudouin, Fimbul, Amery and Larsen D ice shelves, respectively. The formation of ice slabs thus increases the divergence between firn air depletion for the high-end warming scenario on the one hand, and for strong and intermediate mitigation scenarios on the other hand.

Our results suggest that extreme melt seasons can initiate ice slab formation (Fig. 9), in line with current ice slab formation in Greenland (Culberg et al., 2021). When ice layers that formed during an extreme melt season remain in contact with the surface hydrology, they concentrate new refreezing above their horizon amplifying the contribution of even average subsequent melt seasons to ice slab formation. However, standing water on top of refreezing layers is not considered by IMAU-FDM, but an alternative positive feedback mechanism is captured: denser firn and ice has a higher thermal conductivity, which contributes to

more efficient conductive cooling and thereby promotes the growth of ice layers and slabs (Vandecrux et al., 2020). The future accessible FAC on these ice shelves will depend on the frequency of future extreme melt events, and the timing of depletion is therefore less certain compared to non-ice slab regions where the depletion is mainly forced by the mean climate state and therefore more gradual.

Ice shelves where ice slab formation under SSP5-8.5 is prevented have current accumulation rates of $> 600\ \mathrm{mm\,yr^{-1}}$ and

mean annual temperatures of $>$ -19 °C, and are mainly situated in the Antarctic Peninsula and West-Antarctica, such as Wilkins, Stange, Abbot, Crosson and Getz ice shelves. This implies that high accumulation allows new pore space to regenerate FAC above the most recent refreezing layers. In addition, mild temperatures and more low-density fresh snow, both reducing the refreezing capacity per $\mathrm{m^3}$ of firn, prevent firn to reach the ice density. The projected increasing snowfall in Antarctica is thus not only important to counteract ice-sheet mass loss, but also to prevent ice slab formation. This is in line with MacFerrin

et al. (2019), who found that ice slabs appear to be absent in regions of high accumulation ($> 572\ \mathrm{mm\,yr^{-1}}$) on the Greenland ice sheet. These results highlight the different response of ice shelves with low- and high accumulation rates to atmospheric warming.

The AIS ice-shelf-wide runoff extent in Fig. 8 (25 % by 2100 for SSP5-8.5) is substantially lower than reported by Gilbert and Kittel (2021) (98 % by 2100 simulated by the regional climate model MAR forced by CESM2 scenario SSP5-8.5). In

MAR, liquid water reaching a firn layer with a density $> 830\ \mathrm{kg\,m^{-3}}$ is converted to runoff, whereas in IMAU-FDM ice layers are completely permeable for meltwater percolation. Both models thus represent extremes in permeability assumptions, which explains the large difference. This also underlines the potentially large impact of ice slabs when estimating the onset of meltwater ponding. Besides this, MAR also yields higher melt rates compared to RACMO2 for the same forcing (Carter et al., 2022), potentially caused by a more active snowmelt-albedo feedback. Differences in the initialization of the firn layer could

also contribute to the discrepancy.

According to our approach, extensive meltwater ponding is expected when accessible FAC is depleted. Figure 8 shows that in general runoff is enhanced when total FAC (averaged over the ice shelf) roughly approaches 2.5 m for low-accumulation ice





shelves, such as Roi Baudouin ice shelf, and 5 m, for high-accumulation ice shelves, such as Wilkins ice shelf. We expect that the accessible FAC thresholds for runoff generation are approximately similar. Based on these assumptions, Fig. 8 indicates

extensive melt ponding onset around 2060 on Roi Baudouin ice shelf for all scenarios. On Larsen C and Wilkins ice shelves meltwater ponding is expected around 2060 for SSP5-8.5 and around 2080 for SSP1-2.6 and SSP2-4.5. For these three ice shelves, meltwater ponding is thus expected to occur in the 21st century irrespective of the scenario. For SSP5-8.5 extensive meltwater ponding is expected to occur on George VI around 2070, on Fimbul around 2075, and on Shackleton, Amery and Pine Island around 2090. However, extensive meltwater ponding has already been observed on the northern George VI ice shelf

(van Wessem et al., 2023), which suggests that our initial FAC estimates (Fig. 2) here are overestimated. On the other hand, locally observed meltwater ponding along the grounding lines of Amery and Roi Baudouin ice shelves is in line with our low initial FAC estimates here.

## 5.2  Limitations

FDM v1.2AD uses the bucket scheme to simulate vertical meltwater movement, which does not allow standing water over

impermeable ice slabs, lateral meltwater movement, or preferential flow. Neglecting these processes has amongst others an impact on densification and ice layer formation. Verjans et al. (2019) found that a firn model using the bucket scheme can produce similar density results as the more physically based Richards equation in a single domain for four locations on the Greenland ice sheet. Using the Richards equation in a dual domain, which accounts for partitioning between matrix and fast preferential flow, simulated more ice layers, underestimates the FAC to a greater extent, but is better at reproducing density

variability with depth. Using two firn models that use the bucket scheme, Thompson-Munson et al. (2023) simulated ice slabs in Greenland, and found that most ice slabs detected by IceBridge accumulation radar data in 2014 (MacFerrin et al., 2019) overlap with simulated ice slabs and ablation zones. On average, the observed ice slabs are located at slightly higher altitudes than the modelled ones. However, the results are also strongly influenced by the densification equation and climatic forcing used (Verjans et al., 2019).

The densification scheme used in FDM v1.2AD (Eq. 8) is developed for dry-snow densification. The presence of liquid water may also impact the densification rate of firn; however due to a lack of physical understanding and available measurements, this effect has not been included. Equation (8) depends on initial grain size (taken here as 0.03 mm), which is a poorly constrained parameter over the range of climates on ice-sheets. Fortunately, the model grain size is not very sensitive to the initial value, since the growth rate is independent of the initial grain size, and the grain size quickly becomes magnitudes larger (e.g. a growth

rate of $0.0056 \, \mathrm{mm \, day^{-1}}$ for a firn layer of 250 K). Observed and modelled albedo evolution through melt-refreezing cycles show that refreezing strongly increases the grain size (e.g., van Dalum et al., 2022). However, tests with a different refreezing grain size (e.g. 0.4 mm instead of 0.25 mm) did not lead to very different results (Fig. S4). Again, this process is poorly constrained by direct observations, hence the impact of refreezing on grain size, and subsequently on firn compaction, remains uncertain. Firn compaction by horizontal divergence and strain softening is not included in the model, therefore densification

in high-strain and horizontal stretching areas, such as Pine Island and Thwaites ice shelves, is likely underestimated (Horlings et al., 2021; Oraschewski and Grinsted, 2022).



Our outcomes of accessible FAC depend on ice slab formation, which is a complex process to model. That is why, the current approach (as a diagnostic post-processing step, rather than including it in the model) should be regarded as an exploratory study. We present these results together with the total FAC and test the sensitivity to several ice layer thickness permeability relations.

We found that the timing and magnitude of accessible FAC depletion averaged over ice shelves is not very sensitive (< 10 years and < 1 m) to the thickness permeability relationships used (Figs. 3 and 8). Including the impermeability of ice layers interactively within IMAU-FDM will be tested in future work.

FDM v1.2AD does not include a liquid-water routing scheme that can transport ponded water laterally by streams and rivers. The current view is that meltwater ponding weakens ice shelves, potentially leading to ice-shelf disintegration by hydrofrac-

turing (Banwell et al., 2013). However, Bell et al. (2017) found that when the ice surface slope is sufficiently steep, rivers can form that transport liquid water laterally and export the meltwater into the ocean, thereby preventing its destructive effects. Therefore, our approach likely overestimates the vulnerability of ice shelves to hydrofracture.

Our results are furthermore limited by using only a single general circulation model (CESM2) and one model for dynamical downscaling (RACMO2). Future work should address this by using an ensemble of Earth system models and other regional

climate models such as MAR (Kittel et al., 2021) and MetUM (Orr et al., 2021), which have different outcomes for melt (Carter et al., 2022). The exact timing and amount of (accessible) FAC depletion will vary when using a different forcing dataset, however the spatial variability among climatic regions and the dependency of ice layer formation on accumulation should not be fundamentally different.

## 6 Conclusions

In this study, we explored possible future (accessible) FAC evolution over the AIS under various emission scenarios. Our main tool is a firn model with an updated densification expression that allows for changing climatic conditions. Compared to the previous model version, the Ross and Filcher-Ronne ice shelves emerge as more vulnerable to FAC depletion, while high-accumulation ice shelves, such as Getz, Wilkins and George VI are less vulnerable. In our simulations, ice shelves in the Antarctic Peninsula and Roi Baudouin ice shelf in Dronning Maud Land are particularly vulnerable to FAC depletion, also un-

der mitigation scenarios SSP1.2-6 and SSP2-4.5 (> 50 % decrease in FAC). In the high-end warming scenario SSP5-8.5 all ice shelves experience significant FAC depletion by 2100 (> 19 % decrease). The formation of near-surface ice slabs further reduces accessible FAC under SSP5-8.5, especially on ice shelves which currently receive less than 500 mm accumulation annually and have significant interannual variability in surface melt. This includes many East-Antarctic ice shelves and Filchner-Ronne, Ross, Pine Island and Larsen C ice shelves. Under mitigation scenarios SSP1-2.6 and SSP2-4.5, ice slab formation is limited to

Roi Baudouin, Fimbul and Amery ice shelves. High accumulation rates (currently > 600 mm yr$^{-1}$) on Antarctic Peninsula and West Antarctic ice shelves, such as Wilkins, George VI, Getz, Abbot, Stange and Crosson ice shelves prevent the formation of ice slabs under all scenarios. These results underline the different response of low- and high-accumulation ice shelves to atmospheric warming, and identify a potentially large impact of ice slab formation on the viability of low-accumulation ice shelves.



*Code availability.* The code of IMAU-FDM v1.2AD is available on Zenodo (https://doi.org/10.5281/zenodo.8390614, Veldhuijsen et al. 2023b).

*Data availability.* The IMAU-FDM v1.2AD simulations (including total firn air content and accessible firn air content) are available on Zenodo (https://doi.org/10.5281/zenodo.8381267, Veldhuijsen et al. 2023c).

*Author contributions.* SV and WJvdB defined the research goals and designed the study. SV updated the model, performed the simulations
and analyzed the results. All authors contributed to discussions on the manuscript.

*Competing interests.* MRvdB is a member of the editorial board of journal The Cryosphere

*Acknowledgements.* This work was funded by the Netherlands Organization for Scientific Research (grant no. OCENW.GROOT.2019.091).
MvdB acknowledges support from NESSC. We acknowledge ECMWF for computational time on their supercomputers.

.



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
