# Peer review of "Evolution of Antarctic firm air content under three future warming scenarios"

_EGUsphere, 2023_

## Referee Comment (RC1)

**Evolution of Antarctic firn air content under three future warming scenarios**

Sanne B.M. Veldhuijsen, Willem Jan van de Berg, Peter Kuipers Munneke, Michiel R. van den Broeke

*Submitted to The Cryosphere*
* * *
**General comments**

Veldhuijsen et al. update the highly used IMAU Firn Densification Model to account for changing climate conditions, apply a new ice slab permeability expression, and ultimately simulate firn air content (FAC) across the Antarctic Ice Sheet in three future warming scenarios. The authors provide a detailed description of the new model configuration as well as an evaluation of the model. This allows them to explore the evolution of FAC in a changing climate with more confidence.

This paper provides valuable information about how Antarctic firn could behave in different warming scenarios, which is critical for accurately quantifying the ice sheet's future contribution to sea level rise and assessing the stability of ice shelves. Moreover, this paper has a clear motivation and provides an important advancement for semi-empirical firn modeling. As such, this research fits within the scope of *The Cryosphere* and will be an asset to the firn modeling community.

While the science, methodology, and key results of this work are well done and valuable, improvements to the presentation of this work need to be made for publication. This paper is somewhat difficult to follow—even as a reader with a background in firn modeling. A lot of information presented feels disconnected from the key findings related to future Antarctic FAC evolution. My primary suggestion (in more detail below) is to remain more focused on the applications of the model throughout the paper and reconsider what information is pertinent to the key results that are outlined in the abstract.

**Specific comments**

**Excessive Information in Sections 2 and 3**

This paper represents a phenomenal amount of work, and because of that, it reads as two separate papers in its present form. Sections 2 and 3 contain a detailed description of the model adjustments as well as results of the new methodology. Because these sections have methods, results, and five figures, they feel like their own separate study. It's clear that an evaluation of the new model is needed for the results presented in the rest of the paper to be trustworthy. However, Sections 2 and 3 contain so much information that it's difficult to take away the key points. From the rest of the paper, it seems that the main tool being used to calculate future FAC evolution is IMAU-FDM v1.2AD-C, but four model configurations are listed in Table 2 and evaluated in various combinations (e.g., Fig. 4 shows results of three of the model configurations, and Fig. 5 shows some comparisons but not all).

As a reader, I want to walk away from Sections 2 and 3 knowing (1) why a new configuration was needed, (2) what the new configuration is, and (3) how much it improves upon the existing configuration. Points (1) and (2) were well executed in this work, but (3) is where the study falls short. There's so much additional information about four separate model configurations that the reader cannot easily understand what is being compared and why. In my opinion, these sections distract the reader from the main results concerning the application of the new model configuration.

To remedy this, one suggestion is to rewrite Sections 2 and 3 with a very clear focus on the development, calibration, and evaluation of IMAU-FDM v1.2AD-C. Anything important but not directly pertinent to IMAU-FDM v1.2AD-C could go in the supplement. Using Fig. 4 as an example, it does not seem necessary as a reader to know about the MO ratios and fits of model configurations that are not used to create the key results of the study. Perhaps the IMAU-FDM v1.2AD-C results in Fig. 4 could remain in the main text and the other data could go in the supplement.

Reasoning for Parameter and Methodology Choices

There are a handful of places throughout the study where a parameter or methodology choice is made but the reasoning is not provided or not adequately explained. For example, lines 135–144 contain several values for model parameters but little explanations as to why they were selected. A few other places that could use further reasoning are: the choice to use a sigmoid function (line 218), the choice of applying the permeability factors by multiplying (line 228; is there a mathematical or physical reasoning for why firn permeability would be multiplicatively dependent on the overlying layers?), and the selection of different time periods for analysis. This final point is important since it provides the reader with context as to why certain times are being compared. It's clear that a lot of thought has been put into this research and the choices made were intentional, but without explicitly telling the reader why a method/parameter was chosen, it can seem arbitrary.

Full ice sheet vs ice shelves

It's unclear whether this study is focusing on FAC on the entire ice sheet or just the ice shelves. The abstract and conclusions seem to exclusively focus on the ice shelves, but at least half of the figures are showing results of the entire ice sheet. Perhaps including a few sentences in the abstract/conclusion mentioning changes to the rest of the ice sheet beyond the ice shelves could remedy this.

Also, the limitations (Section 5.2) should mention that the calibration data mostly come from non-ice shelf areas (Fig. 2). We are of course limited by the availability of observations on ice shelves, but it should still be mentioned since there's such a heavy focus on the results over ice shelves.

Consistency in accumulation and melt units – Both "mm w.e. yr$^{-1}$" and "mm yr$^{-1}$" are used throughout this paper. I recommend using just the former if possible.

**Line-by-line comments**

Abstract

7 – Consider specifying what makes these climate scenarios different. Suggestion: "three climate scenarios" → "three climate emissions scenarios"

8 – Since "accessible FAC" has not explicitly been defined yet, I suggest adding a short explanation: "To estimate the accessible FAC" → "To estimate the accessible FAC, which is the pore space meltwater can reach," (or something along those lines).

9-10 – This sentence could be more impactful if the reader knows the timeframe to which these results are referring. > 50 % depletion by what year? Or over how many years?

10 – "strong and intermediate mitigation scenarios" is somewhat vague and unclear. If referring to SSP1-2.6 and SSP2-4.5, perhaps denote that: "strong (SSP1-2.6) and intermediate (SSP2-4.5) mitigation scenarios". Perhaps the lack of clarity comes from the idea that these scenarios are often described using their emission strengths rather than their mitigation strengths. The authors could also reword this as "low (SSP1-2.6) and intermediate (SSP2-4.5) emissions scenarios".

13-15 – This is a great sentence with high impact. A small suggestion is to change the word "viability" to something else. Perhaps "vulnerability", "longevity", "stability", or even "instability" could work instead.
1 Introduction

19 – "Both reduce their buttressing effect" → "Both reduce the buttressing effect of ice shelves"

28-29 – The sentence beginning with "Currently" is a little unclear and could use some rewording to help the reader understand. It's unclear how "(i.e. where sufficient tensile stress is present)" is related to the language used before.

32 – "impermeable ice slabs" → "low-permeability ice slabs"

39 – "While runoff is a measure of firn saturation" feels a like a bit of a leap, especially since firn can be unsaturated at depth but have near-surface ice slabs that drive runoff. Suggestions: "While runoff is indicative of the firn saturation", "While runoff is strongly related to firn saturation", or something along those lines

39-43 – In addition to the above comment, these sentences could use some re-wording or re-organization. Specifically, the sentence "The main advantage of using an offline firn model…" feels very out of place. It's unclear if the sentence before that one was describing the alternative to an offline firn model. Also, the final sentence in this paragraph could be moved to Section 5.2.

46 – "forced by outputs of regional climate models" feels redundant and can probably be removed since it's written verbatim in the previous sentence.

51-53 – Consider splitting the sentence "These ice slabs…" into two sentences since there is a lot of information packed in here and there is a natural break in the information flow before "can impede…"

2 Methods

74 – "referred as" → "referred to as"

74-76 – If possible, it could be helpful to briefly summarize the findings from the evaluation in Veldhuijsen et al. (2023).

89 – Should the accumulate rate units instead be "kg m$^{-2}$ s$^{-1}$"? It seems that the "-1" superscript is missing for the seconds.

98-101 – The description of the variable $D$ is unclear and needs to be rephrased to help the reader understand. Is it saying that $D = 0.03$ for $\rho > 550$ kg m$^{-3}$, and $D = 0.07$ for $\rho < 550$ kg m$^{-3}$? Also, splitting this sentence into two could help with clarity and flow.

132 (Eq. 9) – It's unclear how this equation represents the local long-term mean accumulate rate. The units of the variables in this equation have not been explicitly stated, but pressure typically has units of kg m$^{-1}$ s$^{-2}$ and age should have units of time. Therefore, the accumulate rate from Eq. (9) would have units of kg m$^{-1}$ s$^{-3}$ instead of kg m$^{-2}$ s$^{-1}$. It seems something is either incorrect with the equation or the units, but regardless, a clearer explanation is needed for defining this long-term mean accumulation rate.

142 – What is the significant of the thickness range? It's also unclear how the layer thickness explicitly affects the densification of the freshly fallen snow.

Figure 1 – The use of shared *y*-axes makes this figure feel overly complicated. The language used for the melt and accumulation axis label makes its meaning ambiguous (i.e., it seems like it's saying melt *plus* accumulation when that is not the case). An alternative label could be "mass flux", or it could be "Surface melt (mm w.e. yr$^{-1}$) [line break] accumulation (mm w.e. yr$^{-1}$)", or just something to convey that these aren't added together. The fact that the temperature axis spans -45 to -15 °C is somewhat misleading as

well since the temperatures actually only span ~-39 to -29 °C. Finally, having temperature in between accumulation and melt is confusing since accumulation and melt share a *y*-axis. Please note, most of these are just suggestions that would make the figure easier to interpret, but they are not absolutely essential to change.

192 – Please specific which version(s) of the firn model is being compared to the observations. Based on Fig. 2, it seems that it's FDM v1.2AD-C, but no explanation is given as to why that version is being used here.

Figure 2 – Consider changing the color and/or of the star since yellow does not stand out with that color map. Cyan or magenta may work better.

206 – "3–5 cm" → "0.03–0.05 m"

203-207 – This argument was hard to follow and could benefit from clearer language. It's not immediately apparent whether the results of Samimi et al. (2020), (2021), and Clerx et al. (2022) are in agreement or not.

211 – This final phrase is confusing; please elaborate or clarify.

215 – It looks like the citation has not been added in the correct part of the sentence.

Figure 3 – The use of an inverted *y*-axis for FAC is not intuitive. A quick glance at this figure makes it seem that density is increasing and FAC is also increasing.

3 Calibration and model performance

Figure 4 – Should "FDM v1.2A-C" instead be "FDM v1.2AD-C" in the caption?

244 – What is the "($< z_{550}$)" referring to?

Figure 5 – Flip "(e)" and "(f)" in caption. It could be useful to explicitly state how the difference in (d) and (f) is calculated, or at least say "positive values indicate greater FAC due to x model, negative values indicate greater FAC due to y model". Also, why does panel (e) only show two of the models? Why not also include FDMv1.2AD-C?

4 Results

273-274 – What is meant by "most ice shelves"? Are the values reported not referring to all ice shelves?

274-279 – Should all of these percentages have negative signs? It seems that the sign of the change is the same for all but only some are reported as negative.

287 – It seems another negative sign is missing on "76 %".

292-293 – The sentence, "For 42 % of those…" could benefit from being reworded. It's also unclear why a range of years is reported rather than a single year. Another sentence at the beginning of the paragraph could help set up the reader to understand why that range is being evaluated.

294 – There is an inconsistency between the date range in Fig. 7 (2005-2014) versus what's reported here (2005-2015).

302-305 – In the description of Fig. 7c, it could be useful to mention that differences between total and accessible FAC are being calculated over 2060-2100 because that is (presumably?) when they begin diverging. Additionally, please note either here or in the Fig. 7 caption how the difference is calculated. Is it the mean over the 2060-2100 period, or the difference in the final FAC values at 2100, or the time-integrated difference?

328 – "arial" → "areal"

329 – What is meant by "only limited meltwater storage capacity left"? Is there some kind of threshold prescribed here?

Figure 8 – Please consider making the gray grid lines lighter or thinner so as not to distract from the actual data. If possible, please make the lines in the legend thicker so they are easier to see. This figure has some really important information but it's difficult to visualize in its current presentation.

348-351 – Consider rewording or splitting up this sentence to make it easier for the reader to follow.

Figure 9 – The description of panels (a) and (b) in the caption is vague. It needs to be clarified (as it was in the main text in line 345) that one is showing absolute and one is relative. Also, why is the 25 % threshold applied here? Is there some reason why not all areas are shown? As for panels (c-h), there is a lot of information packed in here and the use of so many $y$-axes is hard to follow. The main text primarily discusses MOA (lines 352-360), so perhaps the surface melt and accumulation could be removed from this figure, especially since they are used to calculate MOA. If those variables are retained, check that the units are correct (should they be mm yr$^{-1}$ or mm w.e. yr$^{-1}$?)

363 – Why this date range?

378 – Change units of K to °C to remain consistent.

Figure 10 – Would it not be more useful to see a 1:1 comparison of the two models? In other words, FAC from FDM v1.2A vs FAC from FDM v1.2AD in panel (a)? This is just a suggestion and can be ignored.

5 Discussion

406-408 – Citation needed for this sentence: Jullien et al., 2023 perhaps.

423-424 – Consider rewording this to make it easier to read.

458-460 – Has this sensitivity been tested and reported somewhere?

**References used in this review**

Jullien, N., Tedstone, A.J., Machguth, H., Karlsson, N.B., Helm, V.: Greenland Ice Sheet Ice Slab Expansion and Thickening, Geophysical Research Letters, https://doi.org/10.1029/2022GL100911, 2023.

---

## Author Response (AR1)

**Response to Reviewer 1**

First of all, we would like to thank the reviewer for their time for reviewing and editing our work. We appreciate the constructive feedback we received. Following the feedback we have clarified and revised the structure of the text. We believe that the given suggestions will improve the presentation of our results and improve the readability. Responses to the comments of the reviewers are written in **red** and citations of the manuscript are written in **blue**.

Kind regards, Sanne Veldhuijsen

**General comments**

This paper provides valuable information about how Antarctic firn could behave in different warming scenarios, which is critical for accurately quantifying the ice sheet's future contribution to sea level rise and assessing the stability of ice shelves. Moreover, this paper has a clear motivation and provides an important advancement for semi-empirical firn modeling. As such, this research fits within the scope of The Cryosphere and will be an asset to the firn modeling community. While the science, methodology, and key results of this work are well done and valuable, improvements to the presentation of this work need to be made for publication. This paper is somewhat difficult to follow—even as a reader with a background in firn modeling. A lot of information presented feels disconnected from the key findings related to future Antarctic FAC evolution. My primary suggestion (in more detail below) is to remain more focused on the applications of the model throughout the paper and reconsider what information is pertinent to the key results that are outlined in the abstract.

**Specific comments**

*Excessive Information in Sections 2 and 3*

This paper represents a phenomenal amount of work, and because of that, it reads as two separate papers in its present form. Sections 2 and 3 contain a detailed description of the model adjustments as well as results of the new methodology. Because these sections have methods, results, and five figures, they feel like their own separate study. It's clear that an evaluation of the new model is needed for the results presented in the rest of the paper to be trustworthy. However, Sections 2 and 3 contain so much information that it's difficult to take away the key points. From the rest of the paper, it seems that the main tool being used to calculate future FAC evolution is IMAU-FDM v1.2AD-C, but four model configurations are listed in Table 2 and evaluated in various combinations (e.g., Fig. 4 shows results of three of the model configurations, and Fig. 5 shows some comparisons but not all).

As a reader, I want to walk away from Sections 2 and 3 knowing (1) why a new configuration was needed, (2) what the new configuration is, and (3) how much it improves upon the existing configuration. Points (1) and (2) were well executed in this work, but (3) is where the study falls short. There's so much additional information about four separate model configurations that the reader cannot easily understand what is being compared and why. In my opinion, these sections distract the reader from the main results concerning the application of the new model configuration.

To remedy this, one suggestion is to rewrite Sections 2 and 3 with a very clear focus on the development, calibration, and evaluation of IMAU-FDM v1.2AD-C. Anything important but not directly pertinent to IMAU-FDM v1.2AD-C could go in the supplement. Using Fig. 4 as an example, it does not seem necessary as a reader to know about the MO ratios and fits of model configurations that are not used to create the key results of the study. Perhaps the IMAU-FDM v1.2AD-C results in Fig. 4 could remain in the main text and the other data could go in the supplement.

Thank you for these comments. We agree that the paper contains a lot of information, and we can imagine that it can therefore be difficult to follow. We have rewritten Section 2 and focus on the development and evaluation of FDM v1.2AD-C and thereby we have omitted and moved some redundant information to the supplementary material. E.g.:
- Description of thermodynamics in IMAU-FDM
- Description of the general densification expressions of Arthern et al. (2010)
- Information on updates in layer merging/splitting
- Information on model initialization

- Description of changes in 21$^{st}$ century melt, temperature and accumulation in the RACMO-CESM forcing for all scenarios (Including Figure 1).
- Detailed information on the calibration results (Including Figure 4)
- Detailed information on IMAU-FDM differences (Including Figures 5d,e,f)

The structure of Section 2 and 3 is as follows:
2. IMAU-FDM model updates
2.1 Densification expression
2.2 Atmospheric forcing
2.3 Experimental setup
2.4 In situ measurements
2.5 Calibration
2.6 Performance of the dynamical densification model.
3 Parameterisation of accessible firn air content
(see Sections 2 and 3 and the supplement of the revised MS).

Point (3) is not one of the takeaways. We explain with Point (1) why the new configuration is needed. However, for the historical period, which we need to use for evaluation, the performance of the models is comparable, which is also logical, since there are no significant trends in temperature and accumulation. We have added in Section 2.6: "Overall, the results of the (non-)dynamical versions are similar over the historical period, which is as expected since there are no trends in accumulation and temperature, reducing the impact of the dynamical densification formulation."

*Reasoning for Parameter and Methodology Choices*
There are a handful of places throughout the study where a parameter or methodology choice is made but the reasoning is not provided or not adequately explained. For example, lines 135–144 contain several values for model parameters but little explanations as to why they were selected.
The FDM v1.2A merging and splitting boundaries already existed and were not chosen in this study. To approximate densification of freshly fallen snow, we keep the upper layer to be 1 cm instead of 15 cm (boundaries 0.8 and 1.2 cm – to avoid layer adjustments each timestep), we expand our explanation: "Accumulation is low (<100 mm w.e. yr$^{-1}$) in most of the AIS and therefore 0.15 m of snow can consist of snow from multiple years." and: "We use this small range around 0.01 m to avoid mass needing to be constantly added or removed to the upper layer to keep it exactly at 0.01 m thickness." (This paragraph has been moved to the Text S1).

A few other places that could use further reasoning are: the choice to use a sigmoid function (line 218), We clarify this in the revised MS: "We use a sigmoid function because it has a characteristic S-shaped curve, representing that at a large spatial scale thin ice lenses are permeable, while even very thick ice layers only approach complete impermeability."

the choice of applying the permeability factors by multiplying (line 228; is there a mathematical or physical reasoning for why firn permeability would be multiplicatively dependent on the overlying layers?), Yes, if an ice slab has a permeability factor of 0.5, it means that only 50% of the water can percolate through. If there are two ice slabs both with a permeability factor 0.5 overlying a specific firn layer. It means that only 25% of the water can reach that specific firn layer. So, you have to multiply it by 0.5 twice.

and the selection of different time periods for analysis. This final point is important since it provides the reader with context as to why certain times are being compared. It's clear that a lot of thought has been put into this research and the choices made were intentional, but without explicitly telling the reader why a method/parameter was chosen, it can seem arbitrary. We agree that it is important to explain the periods which we use for analysis. We add this in Section 2.6: "The average absolute difference in FAC over the AIS between FDM v1.2AD-C and FDM v1.2A-E for the period 1979-2014 (the overlapping historical period) is 5.2 %". And see our responses to comments L363, L292-293 and L302-305.

*Full ice sheet vs ice shelves*

It's unclear whether this study is focusing on FAC on the entire ice sheet or just the ice shelves. The abstract and conclusions seem to exclusively focus on the ice shelves, but at least half of the figures are showing results of the entire ice sheet. Perhaps including a few sentences in the abstract/conclusion mentioning changes to the rest of the ice sheet beyond the ice shelves could remedy this. We agree that this is confusing, to remedy this we decided to change our title to: "Firn air content changes on Antarctic ice shelves under three future warming scenarios." We still show the figures containing the entire AIS, as it would also be difficult to only show the ice shelves area, so therefore it does not take up more space to include the grounded ice. In addition, this gives more insight into the how firn responds to a changing climate.

Also, the limitations (Section 5.2) should mention that the calibration data mostly come from non-ice shelf areas (Fig. 2). We are of course limited by the availability of observations on ice shelves, but it should still be mentioned since there's such a heavy focus on the results over ice shelves. Consistency in accumulation and melt units – Both "mm w.e. yr-1" and "mm yr-1" are used throughout this paper. I recommend using just the former if possible. That is indeed true. We have added this in the discussion section as follows: "In line with this, the in situ observations used for calibration and evaluation of the model are mainly situated in non-melt regions." In addition, we will use mm w.e. yr-1 consistently throughout the MS.

**Line-by-line comments**

Abstract

7 – Consider specifying what makes these climate scenarios different. Suggestion: "three climate scenarios" → "three climate emissions scenarios". Done

8 – Since "accessible FAC" has not explicitly been defined yet, I suggest adding a short explanation: "To estimate the accessible FAC" → "To estimate the accessible FAC, which is the pore space meltwater can reach," (or something along those lines). It is defined in L6: "accessible FAC (i.e. the pore space that meltwater can reach)".

9-10 – This sentence could be more impactful if the reader knows the timeframe to which these results are referring. > 50 % depletion by what year? Or over how many years? Thank you for noticing this, we have included "by 2100".

10 – "strong and intermediate mitigation scenarios" is somewhat vague and unclear. If referring to SSP1-2.6 and SSP2-4.5, perhaps denote that: "strong (SSP1-2.6) and intermediate (SSP2-4.5) mitigation scenarios". Perhaps the lack of clarity comes from the idea that these scenarios are often described using their emission strengths rather than their mitigation strengths. The authors could also reword this as "low (SSP1-2.6) and intermediate (SSP2-4.5) emissions scenarios". We have reworded this by "low (SSP1-2.6) and intermediate (SSP2-4.5) emissions scenarios". We also changed this in other parts of the MS.

13-15 – This is a great sentence with high impact. A small suggestion is to change the word "viability" to something else. Perhaps "vulnerability", "longevity", "stability", or even "instability" could work instead. We change this to "vulnerability".

**1 Introduction**

19 – "Both reduce their buttressing effect" → "Both reduce the buttressing effect of ice shelves" Done.

28-29 – The sentence beginning with "Currently" is a little unclear and could use some rewording to help the reader understand. It's unclear how "(i.e. where sufficient tensile stress is present)" is related to the language used before. To clarify, we rephrase this as: "Currently, 60 % of the ice shelves (by

area) both buttress upstream ice and are vulnerable to hydrofracturing if inundated by meltwater (i.e. where sufficient tensile stress is present) (Lai et al. 2020)."

32 – "impermeable ice slabs" → "low-permeability ice slabs" We agree and have also adjusted this in the abstract.

39 – "While runoff is a measure of firn saturation" feels a like a bit of a leap, especially since firn can be unsaturated at depth but have near-surface ice slabs that drive runoff. Suggestions: "While runoff is indicative of the firn saturation", "While runoff is strongly related to firn saturation", or something along those lines We agree, that runoff is "indicative" instead of "a measure" is better. However, we have rewritten this paragraph, and this sentence is not included anymore. See comment below.

39-43 – In addition to the above comment, these sentences could use some re-wording or re-organization. Specifically, the sentence "The main advantage of using an offline firn model…" feels very out of place. It's unclear if the sentence before that one was describing the alternative to an offline firn model. Also, the final sentence in this paragraph could be moved to Section 5.2. We agree that this part is unclear. These sentences will be separated from the previous ones, and rewritten as an entire new paragraph: "Offline firn models forced by output of regional climate models are useful tools to simulate the transient evolution of the firn layer, and can therefore also be used to assess meltwater ponding onset. The main advantage of using a firn model instead of a climate model is the lower computational cost, which enables a higher vertical resolution, a proper initialization of the firn layer and more extensive sensitivity tests. The disadvantage of using an offline firn model is that interaction with the atmosphere is not possible. In contrast to diagnostic studies that use MOA thresholds (e.g., van Wessem et al., 2023), firn models simulate transient changes of the firn layer, thereby accounting for the time it takes to adjust to new climatic conditions. Firn models have been used to simulate the current (1979 - present) AIS firn layer (Gardner et al., 2023; Keenan et al., 2021; Medley et al., 2022; Veldhuijsen et al., 2023a) and its evolution in response to climate change (Ligtenberg et al., 2014; Kuipers Munneke et al., 2014a)"

46 – "forced by outputs of regional climate models" feels redundant and can probably be removed since it's written verbatim in the previous sentence. Done.

51-53 – Consider splitting the sentence "These ice slabs…" into two sentences since there is a lot of information packed in here and there is a natural break in the information flow before "can impede…" We agree and have rewritten this sentence as two separate sentences: "Ice slabs are common in Greenland (MacFerrin et al., 2019; Culberg et al., 2021) and have locally been observed in Antarctica on Larsen C ice shelf (Hubbard et al., 2016). They can impede vertical meltwater percolation to deeper firn, limiting the fraction of the FAC that is accessible for meltwater. "

**2 Methods**
74 – "referred as" → "referred to as" Done

74-76 – If possible, it could be helpful to briefly summarize the findings from the evaluation in Veldhuijsen et al. (2023). Since we are including FDM v1.2A in the evaluation of this work, and to reduce the length of the paper, we decided to not summarize the evaluation.

89 – Should the accumulate rate units instead be "kg m$_{-2}$ s$_{-1}$"? It seems that the "-1" superscript is missing for the seconds. Thank you for noticing this.

98-101 – The description of the variable D is unclear and needs to be rephrased to help the reader understand. Is it saying that D = 0.03 for $\rho$ > 550 kg m$_{-3}$, and D = 0.07 for $\rho$ < 550 kg m$_{-3}$? Also, splitting this sentence into two could help with clarity and flow.

We have rewritten this sentence in the revised MS: "The constant D has different values above (0.03) and below (0.07) the critical density level of ρ = 550 kg m−3 to represent two distinct densification mechanisms (Herron and Langway, 1980)".

132 (Eq. 9) – It's unclear how this equation represents the local long-term mean accumulate rate. The units of the variables in this equation have not been explicitly stated, but pressure typically has units of kg m-1 s-2 and age should have units of time. Therefore, the accumulate rate from Eq. (9) would have units of kg m-1 s-3 instead of kg m-2 s-1. It seems something is either incorrect with the equation or the units, but regardless, a clearer explanation is needed for defining this long-term mean accumulation rate. The gravity constant g was missing in Eq. 9 and will be added in the revised MS.

142 – What is the significant of the thickness range? It's also unclear how the layer thickness explicitly affects the densification of the freshly fallen snow. We clarify this as follows: "We use this small range around 0.01 m to avoid mass needing to be constantly added or removed to the upper layer to keep it exactly at 0.01 m thickness." In addition, this entire paragraph has been restructured and moved to the supplementary material (See supplement Text S1).

Figure 1 – The use of shared y-axes makes this figure feel overly complicated. The language used for the melt and accumulation axis label makes its meaning ambiguous (i.e., it seems like it's saying melt plus accumulation when that is not the case). An alternative label could be "mass flux", or it could be "Surface melt (mm w.e. yr-1) [line break] accumulation (mm w.e. yr-1)", or just something to convey that these aren't added together. The fact that the temperature axis spans -45 to -15 °C is somewhat misleading as well since the temperatures actually only span ~-39 to -29 °C. Finally, having temperature in between accumulation and melt is confusing since accumulation and melt share a y-axis. Please note, most of these are just suggestions that would make the figure easier to interpret, but they are not absolutely essential to change. To improve the readability of this figure, we break it in subplots:

[Figure]

192 – Please specific which version(s) of the firn model is being compared to the observations. Based on Fig. 2, it seems that it's FDM v1.2AD-C, but no explanation is given as to why that version is being used here. We add that this is FDM v1.2A-E, FDM v1.2AD-E and FDM v1.2A-C.

Figure 2 – Consider changing the color and/or of the star since yellow does not stand out with that color map. Cyan or magenta may work better. We slightly enlarged the star and used turquoise.

206 – "3–5 cm" → "0.03–0.05 m" Changed.

203-207 – This argument was hard to follow and could benefit from clearer language. It's not immediately apparent whether the results of Samimi et al. (2020), (2021), and Clerx et al. (2022) are in agreement or not. To clarify, we reorder these sentences: "A small-scale field experiment conducted in Greenland shows that ice layers of 0.12 m can still be completely permeable for liquid water (Samimi

et al., 2020). On the other hand, in another small-scale field experiment in Greenland, ice lenses of only 3-5 cm have also been found to be partly impermeable (Clerx et al., 2022). Samimi et al. (2021) assume that ice layers thicker than 0.5 m act as impermeable barriers and prescribe a non-linear decrease in permeability between 0.1 and 0.5 m." In this way it is clear that Samimi and Clerx do not agree, and Samimi et al. (2021) is stated afterwards as it presents permeability observations for a different thickness range.

211 – This final phrase is confusing; please elaborate or clarify.
To clarify we rephrase: "For ice layers to be impermeable on the model-resolved spatial scale, this requires at least a larger thickness than the ones found to be impermeable in the small-scale field experiments. Here, we also assume that lateral connectivity of ice layers increases with ice thickness."

215 – It looks like the citation has not been added in the correct part of the sentence. Corrected.

Figure 3 – The use of an inverted y-axis for FAC is not intuitive. A quick glance at this figure makes it seem that density is increasing and FAC is also increasing. We invert the y-axis for FAC:

[Figure]

**3 Calibration and model performance**
Figure 4 – Should "FDM v1.2A-C" instead be "FDM v1.2AD-C" in the caption? Thank you for noticing this, corrected.

244 – What is the "($< z_{550}$)" referring to? $Z550$ is defined in L249, the depth of critical density level rho = 550 kg/m3. For clarity, we rephrased this is "(above $z_{550}$)".

Figure 5 – Flip "(e)" and "(f)" in caption. It could be useful to explicitly state how the difference in (d) and (f) is calculated, or at least say "positive values indicate greater FAC due to x model, negative values indicate greater FAC due to y model".  Also, why does panel (e) only show two of the models? Why not also include FDMv1.2AD-C? (e) and (f) are flipped, and we add in the caption how the difference is calculated: "(FDM v1.2AD-E minus FDM v1.2A-E)".  FDM v1.2AD-C has a different forcing, and here we want to compare the relation between FAC and MOA for the different models with the same forcing (and thus same MOA). As Fig. 5d suggests that there is a relation between those. The panels d-f will be moved to the supplementary material based on other comments by reviewer 1 and 2.

**4 Results**
273-274 – What is meant by "most ice shelves"? Are the values reported not referring to all ice shelves? Yes, they are referring to all ice shelves, so we remove "most".

274-279 – Should all of these percentages have negative signs? It seems that the sign of the change is the same for all but only some are reported as negative. Thank you for noticing this, all have negative signs now.

287 – It seems another negative sign is missing on "76 %". Added.

292-293 – The sentence, "For 42 % of those…" could benefit from being reworded. It's also unclear why a range of years is reported rather than a single year. Another sentence at the beginning of the paragraph could help set up the reader to understand why that range is being evaluated. We rephrase this as follows: "In 42 % of those melt-free locations, FAC decreases by 2100, while in the remaining

58 %, FAC increases." In the revised MS we consider the last year of the historical and future period instead of decades.

294 – There is an inconsistency between the date range in Fig. 7 (2005-2014) versus what's reported here (2005-2015). This should be 2014.

302-305 – In the description of Fig. 7c, it could be useful to mention that differences between total and accessible FAC are being calculated over 2060-2100 because that is (presumably?) when they begin diverging. Additionally, please note either here or in the Fig. 7 caption how the difference is calculated. Is it the mean over the 2060-2100 period, or the difference in the final FAC values at 2100, or the time integrated difference? Thank you for these suggestions, we add in the caption how the difference is calculated and explain in the text why we select the 2060-2100 time period, which is indeed because ice slabs become more widespread after that: "We select this period as this is when total and accessible FAC start to diverge."

328 – "arial" → "areal" Changed.

329 – What is meant by "only limited meltwater storage capacity left"? Is there some kind of threshold prescribed here? We explain now in the method section how runoff is calculated: "Once meltwater has saturated the lowermost firn layer beyond the maximum irreducible water content, we assume that it will leave the firn column as runoff instantaneously."

Figure 8 – Please consider making the gray grid lines lighter or thinner so as not to distract from the actual data. If possible, please make the lines in the legend thicker so they are easier to see. This figure has some really important information but it's difficult to visualize in its current presentation. Thank you, we follow these suggestions. See Fig. 6 in the revised MS.

348-351 – Consider rewording or splitting up this sentence to make it easier for the reader to follow. We rephrase this as: "The differences are lowest in the Bellingshausen Sea region, on ice shelves as Wilkins, George VI and Stange and Abbot, and on the Getz and Crosson ice shelves (< 3.9 m and < 47 %), which are among the warmest and wettest ice shelves of the AIS (> -19 °C and > 600 mm yr−1 ). The absence of ice slabs under these conditions is also depicted in Fig. 5c."

Figure 9 – The description of panels (a) and (b) in the caption is vague. It needs to be clarified (as it was in the main text in line 345) that one is showing absolute and one is relative. Also, why is the 25 % threshold applied here? Is there some reason why not all areas are shown? As for panels (c-h), there is a lot of information packed in here and the use of so many y-axes is hard to follow. The main text primarily discusses MOA (lines 352-360), so perhaps the surface melt and accumulation could be removed from this figure, especially since they are used to calculate MOA. If those variables are retained, check that the units are correct (should they be mm yr-1 or mm w.e. yr-1?) We rephrase this as: "The maximum (a) absolute and (b) corresponding relative difference .." The threshold is applied here, because if there is only a very small increase, a relative increase of 100% does not say much. We change the units to mm w.e. in the figure. The absolute values of accumulation are also discussed in the text, therefore we decided to keep the variables as is.

363 – Why this date range? Here, we include two decades as "end of the century", because from 2090 the firn has already disappeared in both models in quite some regions (See Figure 8/Figure 6 of revised MS). We add this: "We select the period 2080-2100 as the firn layer disappears in 2090 on some ice shelves (Fig. 6b)."

378 – Change units of K to °C to remain consistent. Done

Figure 10 – Would it not be more useful to see a 1:1 comparison of the two models? In other words, FAC from FDM v1.2A vs FAC from FDM v1.2AD in panel (a)? This is just a suggestion and can be

ignored. We used this suggestion in draft versions of this manuscript. However, in a 1:1 comparison it is hard to see differences as all data is close to the 1:1-line.

[Figure]

**5 Discussion**
406-408 – Citation needed for this sentence: Jullien et al., 2023 perhaps. We added this citation.

423-424 – Consider rewording this to make it easier to read. We rephrase this as: "The AIS ice-shelf-wide runoff extent in Fig. 8, 25 % by 2100 for SSP5-8.5, is substantially lower than reported by Gilbert and Kittel (2021), who estimate a runoff extent of 98 % by 2100 using the regional climate model MAR forced by exactly the same CESM2 realisation."

458-460 – Has this sensitivity been tested and reported somewhere? No, this follows from the values mentioned, and therefore we think that presenting testing results is not necessary.

**References used in this review**
Jullien, N., Tedstone, A.J., Machguth, H., Karlsson, N.B., Helm, V.: Greenland Ice Sheet Ice Slab Expansion and Thickening, Geophysical Research Letters, https://doi.org/10.1029/2022GL100911, 2023.

**Response to Reviewer 2**

First of all, we would like to thank the reviewer for their time for reviewing this study. We appreciate the insightful feedback we received, which helps us to improve the readability and structure of the text, and to refine the presentation and interpretation of our results. Responses to the comments of the reviewers are written in **red** and citations of the manuscript are written in **blue**.

Kind regards, Sanne Veldhuijsen

This paper presents important new advances, both in terms of the firn modeling methods and the study of ice slab impacts on ice shelf hydrology. It particularly provides a valuable assessment of how firn meltwater storage will evolve in the 21$^{st}$ century that comes closer than past studies to being grounded in our current understanding refreezing processes in firn. Overall, the work seems to be technically correct and well-reasoned. However, the paper can be a fit hard to follow at times due to the huge amount of work and information that went into the results. The authors could consider streamlining the main text and moving some of the model development work to the supplementary text.

**Major Comments:**

**[1]** I found the organization of Section 2-3 somewhat hard to follow, I think in part because the classic methods-results-discussion format falls short when trying to present what is the equivalent of almost two papers worth of work. You might consider first presenting a streamlined discussion of the model updates, calibration, and performance as one section. The goal of this section would be that the reader finishes it convinced that the FDM v1.2A-C run is a reasonable estimate of the future firn evolution. Right now, this point gets somewhat lost because the text is constantly bouncing back and forth between different experiments. Once this is established, then you can introduce how you will use FDM v1.2AD-C to study FAC across Antarctica. From there, it flows more naturally into the results that really only focus on FDM v1.2AD-C simulations. My suggestion for organization would be something like:

Section 2 – IMAU FDM Model Updates

2.1 – Densification expression

2.2 – Calibration Experiments and Atmospheric Forcing (combined since the experiments are just based on the different forcings)

2.3 – Calibration and Validation Data (aka firn cores)

2.4 – Calibration and Performance of FDM1.2AD-E (demonstrate that dynamic densification is reasonable using historical period)

2.5 – Performance of FDM1.2AD-C (demonstrate that future projections are reasonable)

Section 3 – Experimental setup for the future firn evolution and calculation of accessible firn air content

Section 4 – as it is now

I think that much of the current Sections 2&3 can also be streamlined to focus just on the key information needed to explain to the reader how the dynamical densification is achieved, and that the new model produces believable results for the rest of the 21st century. Maybe some of the detailed

discussion of calibration results and extra figures for intermediate steps like FDM1.2AD-E could be moved to the supplement.

Thank you for these comments and suggestions. We agree that the paper contains a lot of information, and we can imagine that it can be difficult to follow. To improve this, we followed your suggestion by presenting all information of the model development together in Section 2 and the accessible FAC calculation description in Section 3. We have rewritten Section 2 and focus on the development and evaluation of FDM v1.2AD-C and thereby we have omitted and moved some redundant information to the supplementary material. E.g.:
- Description of thermodynamics in IMAU-FDM
- Description of the general densification expressions of Arthern et al. (2010)
- Information on updates in layer merging/splitting
- Information on model initialization
- Description of changes in 21$^{st}$ century melt, temperature and accumulation in the RACMO-CESM forcing for all scenarios (Including Figure 1).
- Detailed information on the calibration results (Including Figure 4)
- Detailed information on IMAU-FDM differences (Including Figures 5d,e,f)

The structure of Section 2 is as follows:
2. IMAU-FDM model updates
2.1 Densification expression
2.2 Atmospheric forcing
2.3 Experimental setup
2.4 In situ measurements
2.5 Calibration
2.6 Performance of the dynamical densification model.

(see Sections 2 and 3 and the supplement of the revised MS).

**Minor Comments:**

**Title:** Perhaps should mention ice shelves, since the paper seems to almost entirely focused on discussing FAC change on ice shelves? That is indeed a good idea. Our new title is: "Firn air content changes on Antarctic ice shelves under three future warming scenarios."

**Line 11:** Choose to describe the future climate scenarios either in terms of mitigation or in terms of emissions, but not both. It is quite confusing to keep track of whether "strong" means "strong emissions" under SSP8.5 or "strong mitigation" under SSP2.6. We agree and have decided to only use: "low, intermediate and high emission scenarios" to avoid confusion.

**Line 159:** Not clear why this is done? Maybe this would be better discussed in the experimental setup section? This is discussed in the new Section 2.3: Experimental setup. "FDM v1.2AD-C, the dynamical model indirectly forced by CESM2, is used to simulate future firn evolution over the AIS. FDM v1.2A-E and FDM v1.2AD-E are used for evaluation of the dynamical model over the current climate (Section 2.4) and FDM v1.2A-C to assess the impact of the dynamical model on future firn evolution (Section 4.5, tested for SSP5-8.5)."

**Line 173 – 183:** This paragraph did not seem particularly enlightening. It just reads like a long list of facts with not much explanation of why they are important, and certainly by the time it becomes relevant in the result, the readers will have forgotten all of this. If any of these statistics are important to the results, I would suggest either bringing them up at that point, or moving to a table and offering a super brief summary of the key points (e.g. accumulation, surface melt, and temperature all increase in the future over all parts of the AIS, with ice shelves seeing the largest increases). We agree, and

followed your suggestions by leaving out these two paragraphs, since they are not directly used in the main results section. In addition, we have moved Figure 1 to the supplement.

**Figure 1:** Something to consider that might help readers visual the different experiments would be label on this plot which experiments map to which time periods with background shading or something. The multiple vertical axes are also confusing. Perhaps just break this up into two plots – one with temperature and one with melt and accumulation. To improve the readability of this figure, we will break it up in subplots, and add a different background shading for the periods. See our response to the comment of reviewer 1 on Figure 1 for the figure.

**Line 198:** Machguth et al. (2016) would be another appropriate citation here. We have included this relevant citation.

**Section 2.5:** A few additional suggestions that can help to bound the range of ice slab permeabilities that you use:

**[1]** Ashmore, D. W., Mair, D. W. F., & Burgess, D. O. (2019). Meltwater percolation, impermeable layer formation and runoff buffering on Devon Ice Cap, Canada. *Journal of Glaciology*, *66*(255), Article 255. https://doi.org/10.1017/jog.2019.80

Modeling studied which showed that an impermeability threshold of 1 m for ice slabs led to the best fit between modeled and measured SMB on the Devon Ice Cap.

**[2]** Charalampidis, C., Van As, D., Colgan, W. T., Fausto, R. S., Macferrin, M., & Machguth, H. (2016). Thermal tracing of retained meltwater in the lower accumulation area of the Southwestern Greenland ice sheet. *Annals of Glaciology*, *57*(72), Article 72. https://doi.org/10.1017/aog.2016.2

Used thermistor measurements to show that no percolation occurred through a 5.5 m thick ice slab at KAN-U even during the summer of 2012.

Note that using the numbers from Culberg et al. (2021) when looking at individual ice slabs is a bit complicated, because what they really show is that a package of many ice lenses that is 1-2 m thick can inhibit percolation to some degree. I think it's okay to use since the numbers are consistent with other papers and a 1-2m layer thickness or impermeability is a conservative take-away, but just good to note that their numbers are not totally comparable to some of these other studies on ice slab thickness. Thank you for these suggestions. We included Reference [2] in the figure (see below this text) as an additional observation and add the following: "In addition, firn temperature measurements in Greenland show that no percolation occurred through a 5.5 m thick ice slab even during an extreme melt year (Charalampidis et a. 2016)." Since Reference [1] is not an observation of actual ice slab thickness and depends on e.g. reanalysis forcing data/model configuration we decided not to include that one in the Figure, to not overcomplicate it. However, an impermeability threshold of 1 m is in line with the other findings. The comment about Culberg et al. 2021 is also how to interpret that study (in combination with Gascon et al. 2013), which we addressed with: "While these large-scale radar observations do not give an exact relation between thickness and permeability they do give an indication that ice layers thicker than 0.5 m are at least partly impermeable on a larger scale." and

"Henceforth, we refer to (a set of) ice layers that have a substantial impact on the accessible FAC as ice slabs."

[Figure]

**Line 232:** It's not clear where the > 900 kgm⁻³ threshold is coming from. Machguth et al. (2016) gives an ice slab density of 873  25 kgm⁻³ in their supplement and Rennermalm et al. (2021) gives a density of 862 kgm⁻³ (Machguth et al., 2016; Rennermalm et al., 2021). 900 kg/m3 comes from our own simulations, where we find that the near-surface refreezing layers generally have this value. If we use a lower threshold this impacts the accessible FAC calculations due to changes in high-density non-refreezing layers in the deep firn, which is not what we are interested in. However, we do acknowledge that layers with a density of >830 kg/m3 are usually defined as being impermeable. We rephrase this as follows: "Impermeable ice layers are usually defined as having a density > 830 kg/m3 (the pore close-off density). Here, we use a threshold of > 900 kg/m3 which corresponds to the density of refreezing ice layers in the model. This choice limits the impact on the accessible FAC of changes in high-density non-refreezing layers in the deep firn."

**Line 273:** I am confused about this statement about "higher FAC". It seems like the preceding sentence says that FAC decreases on average? To clarify this, we have rephrased this as: "This implies that, the effect of enhanced precipitation is weaker than the effect of projected warming."

**Figure 6:** I found it very hard to pick out the differences between the top and bottom row. Perhaps a third row with difference plots between rows 1 and 2 would be valuable. We have added a third row and added this information in the text, to highlight the differences between rows 1 and 2:

[Figure]

**Lines 302 – 305:** how do these thresholds compare to what we know about the climatic conditions for firn aquifers and ice slabs from either Greenland (MacFerrin et al., 2019; Munneke et al., 2014) or the Antarctica Peninsula (Van Wessem et al., 2021)? We compare this to ice slab conditions in Greenland in the discussion: "This is in line with MacFerrin et al. (2019), who found that ice slabs appear to be absent in regions of high accumulation (> 572 mm yr$^{-1}$) on the Greenland ice sheet." To also compare this to firn aquifer conditions, we add: "In contrast, high accumulation (>1000 mm yr-1) and relatively mild (>-19) conditions may lead to the formation of aquifers, which aligns with the absence of extensive refreezing (Munneke et al. 2014; Van Wessem et al., 2021)".

**Figure 7c:** Is the "difference between accessible firn air content and total firn air content" just a subtraction or is it a ratio? It is confusing to me why this would be positive for high melt, moderate accumulation ice shelves where my understanding is that accessible FAC would be lower that total FAC. We clarify in the figure caption how the difference is calculated, and we change the sign, so that low accessible FAC gives a negative value. "(c) Difference between accessible FAC and total FAC (accessible FAC minus total FAC)"

**Lines 328 – 337:** How is runoff defined and calculated in IMAU-FDM? I think this needs to be clarified for the reader. I am not fully convinced that the runoff extent or values are particularly meaningful given the caveat that ice slabs remain permeable in the model. The spatial extent of low accessible FAC seems like a far more valuable metric. How consistent is the spatial extent of low accessible FAC with the spatial extent of runoff as calculated by the model? Thank you for this suggestion, we add an explanation in the methods of how runoff is calculated: "Once meltwater has saturated the lowermost firn layer beyond the maximum irreducible water content, we assume that it will leave the firn column as runoff instantaneously." We agree with your statement that runoff extent is not particularly meaningful given the caveat that ice slabs remain permeable in the model, which we also repeat in L329: "Since ice layers are fully permeable for meltwater percolation in IMAU-FDM, the runoff time series are closely related to the total FAC time series". However, it is difficult to come up with a threshold of what low accessible FAC is (at what accessible FAC the firn gets saturated), as this depends on e.g. the annual melt of a specific region. For example, for the dry and cold Amery ice shelf, an accessible FAC of 3 m may hold all melt water, whereas this might not be the case on a wetter ice shelf such as Abbot. In addition, a certain amount of snowfall accumulates each winter, which also complicates this. To overcome this, we want to incorporate the ice slab thickness permeability relation in the meltwater percolation scheme of the firn model and assess the generated runoff in a forthcoming study, which is mentioned in the discussion: "Including the impermeability of ice layers interactively within IMAU-FDM will be tested in future work."

**Line 338:** This seems surprising for an area that is currently a firn aquifer and is projected to see increased accumulation. Is this runoff coming from FAC depletion or from the bottom of an aquifer? It is indeed coming from the bottom of an aquifer, which we also address in Section 4.3: "In addition, runoff here also occurs year-round from firn aquifers, which are perennial subsurface bodies of liquid water, that become more ubiquitous in a warmer Antarctica (Bell et al., 2018)." We rephrase this in Line 338 as: "For Wilkins ice shelf we see a quick increase from 0 to > 90 % runoff extent for both scenarios, which indicates a limited spatial variation in firn state."

**Line 353:** Please explain how you define an "extreme melt season". Consider marking these seasons in some way on Figure 9. This has been explained: "At Amery, Shackleton and Filchner-Ronne ice shelves, we see that extreme melt seasons can cause a persistent reduction in accessible FAC (indicated by the grey shaded area). We define a melt season as extreme when the melt exceeds the 93 % quantile of the detrended time series." The shaded area has been added in Figure 7c to h.

**Line 414:** Seems appropriate to at least mention something about firn aquifers, though I understand that this is not the focus of this study. We agree and have added a small discussion about firn aquifers at the end of this paragraph: " In contrast, high accumulation (>1,000 mm yr-

[1]) and mild (>-19C) conditions may lead to the formation of aquifers (Kuipers Munneke et al. 2014, Van Wessem et al. 2021), which aligns with the absence of extensive refreezing. Although firn aquifers do not have a depleted (accessible) FAC, the runoff from aquifers have the potential to cause hydrofracturing. Therefore, we will explore the future expansion of aquifers in a forthcoming study."